# Pioneer neutrophils release chromatin within in vivo swarms

Hannah M Isles[1‡], Catherine A Loynes[1], Sultan Alasmari[2§], Fu Chuen Kon[1], Katherine M Henry[1], Anastasia Kadochnikova[3], Jack Hales[1], Clare F Muir[1], Maria-Cristina Keightley[2#], Visakan Kadirkamanathan[3], Noémie Hamilton[1], Graham J Lieschke[2†], Stephen A Renshaw[1†*], Philip M Elks[1†*]

[1]The Bateson Centre and Department of Infection, Immunity and Cardiovascular Disease, University of Sheffield Medical School, Sheffield, United Kingdom; [2]Australian Regenerative Medicine Institute, Monash University, Clayton, Australia; [3]Department of Automatic Control and Systems Engineering, University of Sheffield, Sheffield, United Kingdom

*For correspondence:
s.a.renshaw@sheffield.ac.uk (SAR);
p.elks@sheffield.ac.uk (PME)

†These authors contributed equally to this work

Present address: ‡Division of Hematology/Oncology, Department of Medicine, Weill Cornell Medical College, New York, United States; §Department of Clinical Laboratory Sciences, King Khalid University, Abha, Saudi Arabia; #La Trobe Institute for Molecular Science, Department of Pharmacy and Biomedical Sciences, La Trobe University, Bendigo, Australia

Competing interests: The authors declare that no competing interests exist.

**Abstract** Neutrophils are rapidly recruited to inflammatory sites where their coordinated migration forms clusters, a process termed neutrophil swarming. The factors that modulate early stages of neutrophil swarming are not fully understood, requiring the development of new in vivo models. Using transgenic zebrafish larvae to study endogenous neutrophil migration in a tissue damage model, we demonstrate that neutrophil swarming is a conserved process in zebrafish immunity, sharing essential features with mammalian systems. We show that neutrophil swarms initially develop around an individual pioneer neutrophil. We observed the violent release of extracellular cytoplasmic and nuclear fragments by the pioneer and early swarming neutrophils. By combining in vitro and in vivo approaches to study essential components of neutrophil extracellular traps (NETs), we provide in-depth characterisation and high-resolution imaging of the composition and morphology of these release events. Using a photoconversion approach to track neutrophils within developing swarms, we identify that the fate of swarm-initiating pioneer neutrophils involves extracellular chromatin release and that the key NET components gasdermin, neutrophil elastase, and myeloperoxidase are required for the swarming process. Together our findings demonstrate that release of cellular components by pioneer neutrophils is an initial step in neutrophil swarming at sites of tissue injury.

## Introduction

A robust inflammatory response against invading pathogens or endogenous danger signals requires the coordination of multiple cellular and immune components. Neutrophils are one of the first responders to tissue inflammation and rapidly home to inflamed tissue within hours of injury. Within inflamed tissue, neutrophils destroy pathogens (*Urban et al., 2006*) and clear wound debris (*Wang, 2018*), ultimately leading to the restoration of tissue homeostasis. The anti-microbial repertoire of neutrophils can cause substantial secondary tissue damage and cell death; therefore, neutrophil recruitment to inflammatory sites must be tightly controlled.

Neutrophils are recruited to sites of inflammation through a series of well-defined molecular events during which they are primed by pro-inflammatory signals including growth factors, inflammatory cytokines, and chemoattractants (*Ley et al., 2007*; *Woodfin et al., 2010*; *Nourshargh and Alon, 2014*). Neutrophils are capable of integrating host- and pathogen-derived environmental signals, resulting in their polarisation and migration towards the initiating inflammatory stimulus (*McDonald and Kubes, 2010*). Within the interstitium, neutrophils can rapidly coordinate their migration patterns towards sterile inflammation and infection to form clusters (*Reátegui et al.,*

*2017*; *Lämmermann et al., 2013*; *Ng et al., 2011*; *Chtanova et al., 2008*; *Peters et al., 2008*; *Sun and Shi, 2016*; *Uderhardt et al., 2019*). The parallels between these cellular behaviours and migration patterns seen in insects have led to the use of the term 'neutrophil swarming'.

A series of sequential phases leading to neutrophil swarm formation has been described in murine models: the initial migration of 'pioneer' neutrophils close to the inflammatory site (scouting phase) is followed by large-scale synchronised migration of neutrophils from more distant regions (amplification phase) leading to neutrophil clustering (stabilisation phase) and eventual resolution (*Reátegui et al., 2017*; *Lämmermann et al., 2013*; *Ng et al., 2011*; *Chtanova et al., 2008*). The initial arrest and death of early recruited pioneer neutrophils correlates with the onset of neutrophil swarming (*Lämmermann et al., 2013*; *Ng et al., 2011*; *Uderhardt et al., 2019*) and is mediated by lipid and protein chemoattractants including leukotriene B4 (LTB4) (*Reátegui et al., 2017*; *Lämmermann et al., 2013*). In zebrafish, LTB4 is produced via arachidonic acid metabolism downstream of a sustained calcium wave induced upon neutrophil contact with damage-associated molecular patterns (DAMPs) released by necrotic cells (*Poplimont et al., 2020*). Swarming neutrophils propagate this local calcium 'alarm signal' through connexin-43, Cx43, which amplifies local chemoattractant release and directs migration of neutrophils to form swarms (*Poplimont et al., 2020*). The precise nature of endogenous pioneer neutrophil behaviours and the mechanisms of swarming have yet to be determined in vivo.

At sites of inflammation, neutrophil behaviour can be modulated by extracellular stimuli such as proinflammatory cytokines, pathogens, toxic tissue constituents, and platelets (*Iba et al., 2013*). The final fate of all neutrophils is death, but the mode of cell death can differ depending on stimuli and can show unique macroscopic morphological changes (*Galluzzi et al., 2018*). Controlled cell death, by apoptosis for example, does not cause surrounding tissue damage, whilst uncontrolled neutrophil death, such as necrosis, results in the spilling of cellular contents containing DAMPs (*Poplimont et al., 2020*; *Labbé and Saleh, 2008*). Neutrophils are able to produce extracellular traps (NETs) into surrounding tissues, composed of DNA and histones embedded with granular and cytoplasmic proteins, which are able to capture and kill pathogens extracellularly (*Brinkmann et al., 2004*). Neutrophils release NETs following a series of intracellular changes, resulting in chromatin decondensation, breakdown of the nuclear envelope, and mixing of DNA with granular and cytoplasmic proteins (*Remijsen et al., 2011*). The cellular uptake of propidium iodide and the expulsion of intracellular components from pioneer neutrophils precede the onset of swarming in murine models (*Uderhardt et al., 2019*), suggesting that extracellular DNA release during cell death is a possible mechanism of swarm initiation by pioneer cells. The processes of NET release and neutrophil swarming have been recently demonstrated to be linked, with swarming neutrophils releasing NETs to restrict fungal growth of *Candida albicans* clusters on in vitro arrays (*Hopke et al., 2020*). Further investigation of neutrophil swarming and NET release using in vivo models is required to dissect the molecular mechanisms involved in complex tissue environments.

The zebrafish (*Danio rerio*) is a powerful model organism that has been extensively used to study neutrophil function (*Robertson et al., 2014*; *Loynes et al., 2018*; *Niethammer et al., 2009*). The optical transparency of zebrafish embryos and ease of generating fluorescent transgenic reporter lines allow for the tracking of endogenous neutrophils from the time of injury (*Renshaw et al., 2006*). In this study, we demonstrate using intravital imaging that a single pioneer neutrophil becomes the focal point of migration for swarming neutrophils within damaged tissue. Prior to swarming onset, the pioneer neutrophil adopts a rounded, non-motile morphology distinct from other neutrophils within the inflamed tissue, but is not undergoing apoptosis. Pioneer and early swarming neutrophils release intracellular components including chromatin, into tissues, reminiscent of NET release. We show that neutrophil extracellular chromatin release in zebrafish shares key features of mammalian NETs: they form in response to a range of chemical stimuli and share essential structural features. Inhibition of NET components gasdermin D, neutrophil elastase, and myeloperoxidase is able to reduce the swarming process, indicating an important role for release of nuclear contents from pioneer neutrophils in the swarming response.

## Results

### Neutrophils swarm following injury and infection in vivo

We first investigated neutrophil mobilisation to injury and infectious stimuli. Neutrophil responses to mechanical tissue injury were assessed by tailfin transection of 3 days post-fertilisation (dpf) larvae from the *TgBAC(mpx:GFP)i114* transgenic line (subsequently termed *mpx:GFP*) (*Renshaw et al., 2006*). Neutrophil clusters were observed along the tailfin wound, reminiscent of neutrophil swarming events in mammalian systems (*Figure 1A*, *Video 1*). Infectious stimuli, such as *Staphylococcus aureus*, a gram-positive bacterium, have been shown to cause a robust neutrophil swarming response in mammalian models (*Kamenyeva et al., 2015*). To ascertain the neutrophil swarm response to infection in the zebrafish, we injected *S. aureus* into the zebrafish otic vesicle. We observed robust neutrophil recruitment (21 ± 2 neutrophils) and neutrophil clusters reminiscent of mammalian neutrophil swarming to infection (*Figure 1B*, *Video 2*), which was not seen in larvae injected with a PBS control (1 ± 0.3 neutrophils) (*Figure 1—figure supplement 1A–B*). These data indicate that neutrophil clusters, reminiscent of swarming, are present in response to tissue damage and infection in zebrafish.

Analysis of neutrophil migration patterns within the first 6 hr following tissue injury identified different behaviours of neutrophil clustering over time. In 14% of larva, short-lived, transient neutrophil clusters, which formed and dissipated (stable for <1 hour) multiple times within the imaging period, were observed (*Figure 1—figure supplement 2*, *Video 3*). In the majority of larva (50%), persistent neutrophil clusters were observed, reminiscent of neutrophil swarming reported in mammalian systems (*Reátegui et al., 2017*; *Lämmermann et al., 2013*; *Ng et al., 2011*; *Chtanova et al., 2008*; *Figure 1C*, *Video 1*). We defined a persistent swarm as the formation of a cluster that grew by the coordinated migration of individual neutrophils and continued to grow for at least 1 hr (*Figure 1D*). Persistent swarms began forming from 40 min post-injury (*Figure 1—figure supplement 3A*) and remained stable for an average of 2.17 (±0.32) hr (*Figure 1—figure supplement 3B*). The remaining 36% of larvae showed no evidence of swarming behaviour within the 6 hr imaging period (*Figure 1C*, *Video 3*).

In mammalian neutrophil swarming, biphasic neutrophil responses are modulated by the lipid LTB4 (*Lämmermann et al., 2013*). In the zebrafish tailfin model, two waves of neutrophil recruitment were observed: the early migration of neutrophils proximal to the wound site between 0.5and 2 hpi, followed by a later influx of neutrophils from more distant sites (*Figure 1E*). Biosynthesis of LTB4 in zebrafish occurs through fatty acid metabolism of arachidonic acid, via intermediates found in mammalian systems, resulting in the production of LTB4 by the enzyme leukotriene A4 hydrolase (LTA4H), encoded by the gene *lta4h* (*Poplimont et al., 2020*; *Tobin et al., 2010*; *Chatzopoulou et al., 2016*). Zebrafish have three LTB4 receptors: the high affinity *blt1* receptor and two low affinity receptors *blt2a* and *blt2b*, of which neutrophils predominantly express *blt1* (*Figure 1—figure supplement 4A*). We used the CRISPR-Cas9 system to knock down *lta4h* and *blt1* loci to investigate the requirement for LTB4 in the neutrophil response to a tailfin wound. High-resolution melt (HRM) analysis identified genomic lesions at each locus (*Figure 1—figure supplement 4B*). Early neutrophil recruitment to the wound site at 3 hpi was similar between control (*tyrosinase*, a well-defined CRISPR targeting pigment) (*Isles et al., 2019a*; *Jao et al., 2013*), *blt1* and *lta4h* crRNA injected larvae (*Figure 1F*), suggesting that LTB4 signalling is not required for early neutrophil recruitment. However, by 6 hpi, neutrophil recruitment in *blt1* and *lta4h* crRNA injected larvae was significantly lower than control crRNA-injected larvae (*Figure 1F*), demonstrating a requirement for LTB4 in the second phase of neutrophil recruitment in zebrafish, corroborating findings from mouse (*Lämmermann et al., 2013*) and human (*Reátegui et al., 2017*).

Neutrophil swarms grow by large-scale migration of neutrophils towards early recruited pioneer neutrophils, which release attractant signals, including LTB4 mediated by sustained calcium alarm signals (*Lämmermann et al., 2013*; *Ng et al., 2011*; *Chtanova et al., 2008*; *Poplimont et al., 2020*). We analysed the migration of neutrophils at sites of mechanical injury in the time period preceding swarm formation using our linear tailfin wound model. We identified that all swarms developed by highly directed neutrophil migration towards a single neutrophil rather than randomly occurring at other points along the linear wound (*Figure 2A*). Tracking analysis showed that this individual neutrophil was the focal point of migration (*Figure 2B,C*, *Video 1*). We termed this neutrophil the 'pioneer' as it matched the behaviour of pioneer neutrophils described in murine models

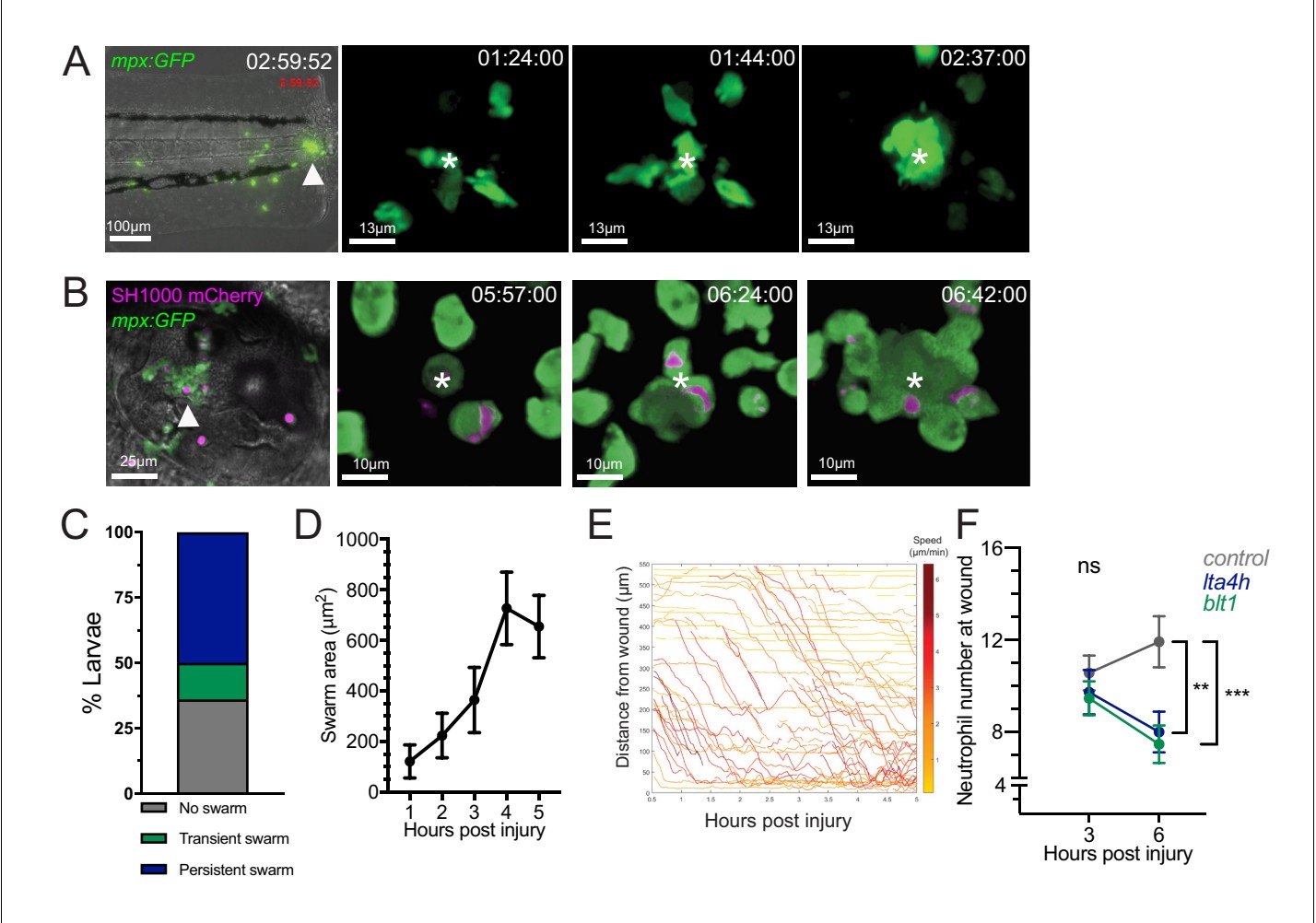

**Figure 1.** Neutrophil swarming occurs as part of the zebrafish inflammatory response. (**A**) Zebrafish neutrophils swarm at sites of tissue damage. Representative image illustrating neutrophils swarming (arrowhead) at the wound site following tail fin transection in 3dpf *mpx:GFP* larvae. Image was taken using 20× magnification on a TE2000U inverted microscope (Nikon). Time stamp shown is relative to the start of the imaging period at 30 min post injury and is h:mm:ss. 3D reconstruction time course illustrating neutrophils swarming at the wound site (swarm centre is highlighted by white asterisk). Imaging was performed using a 40× objective spinning disk confocal microscope (Perkin Elmer). Time stamps shown are relative to time post-injury and are in hh:mm:ss. (**B**) Representative image illustrating neutrophil swarming (arrowhead) in otic vesicle infected with *S. aureus* (magenta). Time stamps shown are hh:mm relative to time post infection. 3D reconstruction time course illustrating neutrophils swarming (swarm centre is highlighted by white asterisk) within the otic vesicle of 2dpf *mpx:GFP* larvae injected with 2500 cfu *S. aureus* SH1000 pMV158mCherry. Imaging was performed using a 20× objective spinning disk confocal microscope. Time stamps shown are hh:mm:ss relative to time post injection. (**C**) The percentage of tailfin transected larvae that had no swarms, transient swarms, or persistent swarms after 6hpi. Data shown are from n = 14 larvae from five biological replicates (*Figure 1—source data 1*). (**D**) Area of neutrophil swarms measured at hourly intervals during the 5 hr imaging period. Error bars shown are mean ± SEM, n = 7 larvae with persistent swarms (*Figure 1—source data 2*). (**E**) Distance time plot demonstrating the early recruitment of neutrophils proximal to the wound site (<350 μm) followed by the later recruitment of more distant neutrophils. Tracks are colour coded based on their average speed (μm/min). (**F**) CRISPR/Cas9-mediated knockdown of LTB4 signalling reduces late neutrophil recruitment. Neutrophil counts at the wound site in control *tyr* crRNA injected larvae (grey line), *lta4h* crRNA injected larvae (blue line), and *blt1* crRNA injected larvae (green line) at 3 and 6 hpi. Error bars shown are mean ± SEM. Groups were analysed using a two-way ANOVA and adjusted using Sidak's multi comparison test. **p<0.008 n = 45 accumulated from three biological repeats (*Figure 1—source data 3*).

The online version of this article includes the following source data and figure supplement(s) for figure 1:

**Source data 1.** Numerical data for the graph of *Figure 1C*.

**Source data 2.** Numerical data for the graph of *Figure 1D*.

**Source data 3.** Numerical data for the graph of *Figure 1F*.

**Figure supplement 1.** Neutrophil migration to the otic vesicle after *Staphylococcus aureus* infection.

**Figure supplement 1—source data 1.** Numerical data for the graph of *Figure 1—figure supplement 1B*.

**Figure supplement 2.** Transient neutrophil swarms at the tailfin wound.

*Figure 1 continued on next page*

(*Lämmermann et al., 2013*; *Ng et al., 2011*; *Lämmermann, 2016*). Prior to swarming, pioneer neutrophils adopted a rounded, non-motile morphology, indicated by their higher circularity index and lower displacement compared to nearest-neighbour neutrophils in the frame immediately preceding the onset of swarming (*Figure 2D,E*).

The swarms which formed around endogenous pioneer neutrophils in zebrafish exhibit similar defined patterns of neutrophil migration to those observed in mammalian swarming (*Video 4*; *Reátegui et al., 2017*; *Ng et al., 2011*). During a scouting phase, neutrophils migrated to the wound site prior to swarm formation over a period which lasted on average 88 ± 24 min (*Figure 2—figure supplement 1A* and analysis of data from *Figure 2A*). Within the scouting neutrophil population, a single pioneer neutrophil adopted the distinct, rounded morphology at the wound site. The initiation phase of swarming began when the pioneer neutrophil stopped migrating and became rounded, marking the site of the swarm centre, and ended when the first neutrophil joined the swarm, taking on average 36 ± 7 min (*Figure 2—figure supplement 1B* and analysis of data from *Figure 2A*). During the aggregation phase, there was a directed migration of neutrophils towards the pioneer to form and consolidate the swarm, for 183 ± 25 min, or until the end of the imaging period (*Figure 2—figure supplement 1B* and analysis of data from *Figure 2A*).

To investigate whether the rounded, non-motile morphology was distinct to pioneers, or common to all neutrophils upon arrival at the wound site, scouting neutrophils (including the pioneer neutrophil) were tracked and their migration patterns analysed for a set time period which covered the scouting and initiation phase (*Figure 2F*). The speed, displacement, and meandering index of pioneer neutrophils were significantly reduced between the scouting and initiation phases. No differences in migration behaviours were observed in neighbouring scouting neutrophils, which migrated to the wound site within the same time period (*Figure 2G*). These data demonstrate that endogenous pioneer neutrophils adopt distinct morphology and migration patterns at the wound site, which is not seen in other wound neutrophils.

## Pioneer neutrophil membranes are intact prior to the onset of swarming

Due to the rounding of pioneer neutrophils prior to the onset of swarming, we hypothesised that these cells may be undergoing a form of programmed cell death. We first investigated their membrane permeability using the DNA intercalating agent, propidium iodide (PI). Pioneer neutrophils excluded PI in 100% of observed swarm initiation events (*Figure 3A–D*, *Video 5*), demonstrating that the plasma membranes of these neutrophils were intact before swarming began. At the same time, pioneer neutrophils were surrounded by dense accumulations of extracellular DNA and cellular debris, confirming that PI was taken up by nearby material in these tissues (*Video 5*). Apoptotic neutrophils can also exclude PI; therefore, a Förster resonance energy transfer (FRET)-based reporter for neutrophil apoptosis (*Tyas et al., 2000*) was used to determine whether pioneer neutrophils were undergoing apoptosis. Analysis of pioneer neutrophils

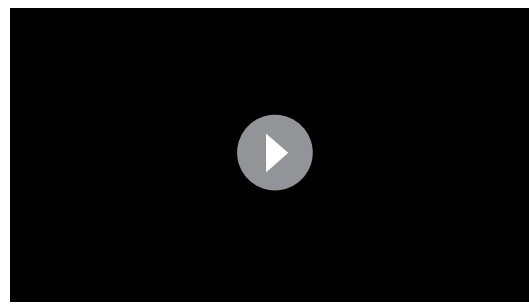

**Video 1.** A pioneer neutrophil stops migrating, rounds up, and becomes the centre of a persistent swarm. A 280 min spinning disk confocal timelapse of a *mpx:GFP* positive neutrophil (filled arrowhead) migrating at the wound site, that stops migrating and adopts a rounded shape (hollow arrowhead), before becoming the centre of a nascent neutrophil swarm (asterisk).
https://elifesciences.org/articles/68755#video1

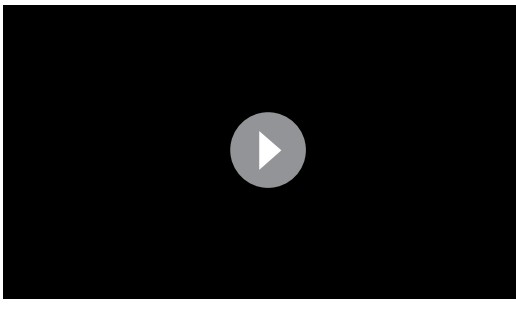

**Video 2.** Neutrophil swarm response to *Staphylococcus aureus*. A 100 min spinning disk confocal timelapse of neutrophils (*mpx:GFP* positive) swarming (asterisk at centre) around *Staphylococcus aureus* (SH1000 pMV158mCherry, red fluorescence) infection in the otic vesicle.
https://elifesciences.org/articles/68755#video2

prior to swarming in *Tg(mpx:CFP-DEVD-YFP) sh237* larvae (*Robertson et al., 2016*) identified that despite the rounded, non-motile morphology, a FRET signal was present during both the scouting and initiation phases in all pioneer neutrophils studied (*Figure 3E*, *Video 5*, n = 6 neutrophils from five experimental repeats), indicating that caspase cleavage did not occur and that pioneer neutrophils were therefore not apoptotic. Neutrophil apoptosis at this early time point during the inflammatory response to the tailfin wound was infrequent; however, on the rare occasion when the neutrophil FRET signal was lost (*Figure 3F*), it was not followed by a swarming response (n = 2 neutrophils). To further confirm that neutrophil apoptosis did not lead to swarming, we assessed neutrophil swarms in the presence of the pan-caspase inhibitor zVAD-fmk. Caspase inhibition did not change the frequency of swarm formation (*Figure 3G*), indicating that apoptosis does not play a significant role in swarm formation. Taken together, these data demonstrate that despite a rounded morphology, pioneer neutrophils exclude PI prior to swarming and are not undergoing apoptosis.

## Swarming neutrophils release cytoplasmic material in balloon-like structures

During the aggregation phase of the swarming response in zebrafish, we identified neutrophil cellular fragments around developing clusters, reminiscent of mammalian neutrophil extracellular trap (NET) release (*van der Linden et al., 2017*; *Tanaka et al., 2014*). Using confocal microscopy, we identified that balloon-like structures, as well as smaller fragments of neutrophil debris, were released from swarms (*Figure 4A*, *Video 6*). This violent release of neutrophil fragments, accompanied by large cytoplasmic structures, has been associated with NET release in mammalian systems (*van der Linden et al., 2017*; *Tanaka et al., 2014*). The cytoplasmic structures observed were larger than the cell body of nearest neighbour neutrophils and the resultant debris (*Figure 4B*). Neutrophil cytoplasmic fragments were produced following a series of striking morphological events: an initial stretching and budding of neutrophil cytoplasm, followed by a violent, catapult-like release of the cytoplasmic structure with eventual formation of associated debris (*Figure 4C,D*, *Video 6*).

We next determined whether the pioneer neutrophil itself could release these striking extracellular structures. Pioneer neutrophils were studied within swarms using a photoconversion approach in a zebrafish reporter line, in which the photoconvertible protein Kaede was expressed specifically in neutrophils: *TgBAC(mpx:GAL4-VP16) sh256;Tg(UAS:kaede)s1999t* (referred to as *mpx:Kaede*) (*Robertson et al., 2014*; *Elks et al., 2011*; *Ellett et al., 2015*). The first neutrophil migrating towards the tailfin wound was photoconverted in multiple larvae, with a small proportion of these going on to become the pioneer of a swarm. Photoconverted neutrophils that went on to become the swarm focus released photoconverted cytoplasmic balloons when they stopped migrating and rounded up, showing that pioneer neutrophils release extracellular material into the surrounding tissue (*Figure 4E,F*). This was confirmed by the cytoplasmic balloon

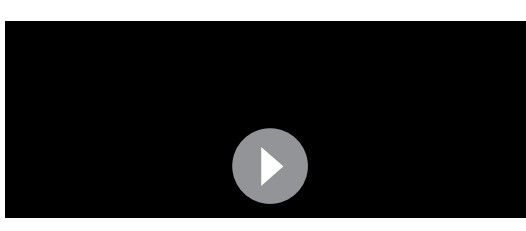

**Video 3.** Transient swarming and no swarming at a tailfin wound. Spinning disk confocal timelapse examples of zebrafish *mpx:GFP* larvae that form transient swarms (white arrowheads) or no swarms at the tailfin wound (to the righthand side of the video).
https://elifesciences.org/articles/68755#video3

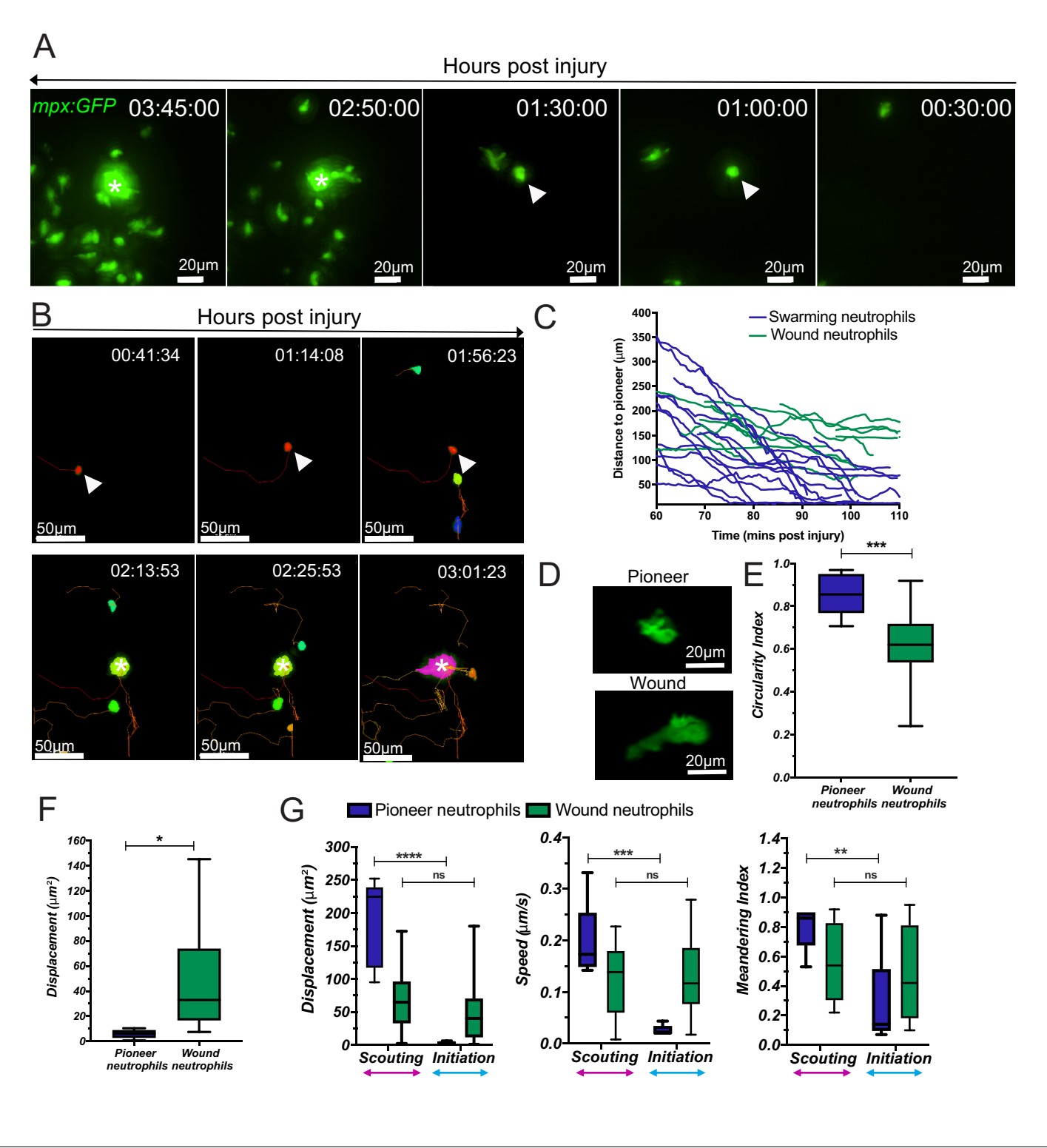

**Figure 2.** Neutrophil swarms develop around an endogenous pioneer neutrophil. (**A**) Reverse chronological time lapse sequence of a persistent neutrophil swarm where one individual neutrophil is visible at the swarm centre (asterisk) prior to neutrophil clustering (arrowhead). Time stamps shown are hh:mm:ss relative to injury time. (**B**) Chronological time lapse sequence of swarming neutrophils. The pioneer is marked with an arrowhead prior to the addition of further neutrophils to the swarm (asterisk). The result of migration is the aggregation of neutrophils to form clusters. (**C**) Distance-time plot (DTP) of individual cell migration paths of neutrophils at the wound relative to the pioneer neutrophil (blue plots are swarming neutrophils and green plots are nearest neighbour wound neutrophils that do not participate in the swarm). Tracks are relative to pioneer neutrophil position; swarming

*Figure 2 continued on next page*

*Figure 2 continued*

neutrophils migrate to the pioneer neutrophil, whilst non-swarming neutrophils do not (n = 4 larvae) (*Figure 2—source data 1*). (D) Representative image of pioneer and non-pioneer neutrophil morphology. Images were taken using a 40X objective lens on a spinning disk confocal microscope (Perkin Elmer). Scale bars are 20 µm. Quantification of pioneer neutrophil migration pattern in the frames preceding swarming. The circularity index (roundness) (E) (*Figure 2—source data 2*) and displacement (movement) (F) (*Figure 2—source data 3*) of pioneer neutrophils and wound neutrophils migrating at the wound site in the same time period (n = 5 larvae, unpaired t-test where *p<0.05 and **p<0.01). (G) Neutrophils were tracked from 30 min post injury. Parameters to study the migration patterns of pioneer and wound neutrophils were compared in the scouting and initiation phases. Neutrophil displacement (the linear distance each neutrophil travelled) (*Figure 2—source data 4*). Neutrophil speed (*Figure 2—source data 5*). Neutrophil meandering index (the displacement divided by the total length of the neutrophil track) (*Figure 2—source data 6*). Error bars are mean ± SEM. Groups were analysed using a two-way ANOVA and adjusted using Sidak's multi comparison test. *p<0.05, **p<0.01, n =5 larvae.

The online version of this article includes the following source data and figure supplement(s) for figure 2:

**Source data 1.** Numerical data for the graph of *Figure 2C*.
**Source data 2.** Numerical data for the graph of *Figure 2E*.
**Source data 3.** Numerical data for the graph of *Figure 2F*.
**Source data 4.** Numerical data for the graph of *Figure 2G* (Displacement).
**Source data 5.** Numerical data for the graph of *Figure 2G* (Speed).
**Source data 6.** Numerical data for the graph of *Figure 2G* (Meandering index).
**Figure supplement 1.** Phases of neutrophil swarming in zebrafish.

---

structure becoming positive for propidium iodide over time, indicating the release of nuclear DNA (*Figure 4G–I*, *Video 6*). Pioneer neutrophil balloon release occurred during the early phases of swarm formation, suggesting that early swarming neutrophils expel DNA.

## Zebrafish neutrophils release extracellular DNA after treatment with known NET stimuli

To determine whether zebrafish NETs are regulated similar to those of mammalian neutrophils, we purified zebrafish neutrophils from adult *Tg(lyz:dsRed)nz50* kidneys and cultured them in vitro, stimulating them with known NET inducers and observing NET production by observation of DNA release using the nuclear dye SYTOX Green (*Figure 5A*). We used chemical stimuli (PMA and calcium ionophore [Cal]) as well as microbial stimuli (*Candida albicans* and *S. aureus*) and showed that all chemical and microbial stimuli resulted in the progressive accumulation of NETs over 120 min compared to control (*Figure 5B*). These data indicate that zebrafish neutrophils can produce NET-like structures in response to a variety of stimuli in vitro.

## Swarming neutrophils release chromatin

To study neutrophil chromatin release in vivo, we generated a transgenic reporter line for neutrophil histone H2az2a (H2A), providing a cell-autonomous reporter of NET release. A genetic construct containing the sequence for H2A with a C-terminal fusion of the fluorescent protein mCherry, driven by the neutrophil-specific *lyz* promoter (*Buchan et al., 2019*; *Hall et al., 2007*), was generated using Gateway cloning (*Figure 6A*; *Kwan et al., 2007*). It was introduced into the genome of *mpx:GFP* larvae by Tol2-mediated transgenesis, and a double transgenic stable line was generated: *TgBAC(mpx:GFP)i114; Tg(lyz:h2az2a-mCherry,cmlc2:GFP)sh530* (sh530 is subsequently referred to as *H2A-mCherry*). The *H2A-mcherry* transgene was expressed by neutrophils in larvae (*Figure 6B*), colocalising with the nuclear stain DAPI (*Figure 6C*). Construct integration did not affect neutrophil migration to sites of inflammation (*Figure 6—figure*

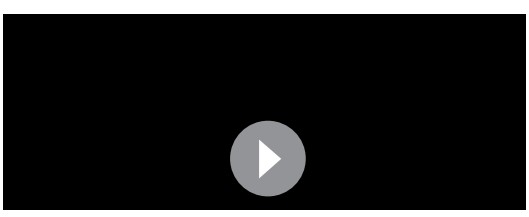

**Video 4.** The scouting, initiation and aggregation phases of neutrophil swarming. Spinning disk confocal timelapse of *mpx:GFP* larva with the brightfield overlaid image shows a developing swarm with scouting, initiation and aggregation phases of swarming labelled. The pioneer neutrophil is indicated by a cell track.
https://elifesciences.org/articles/68755#video4

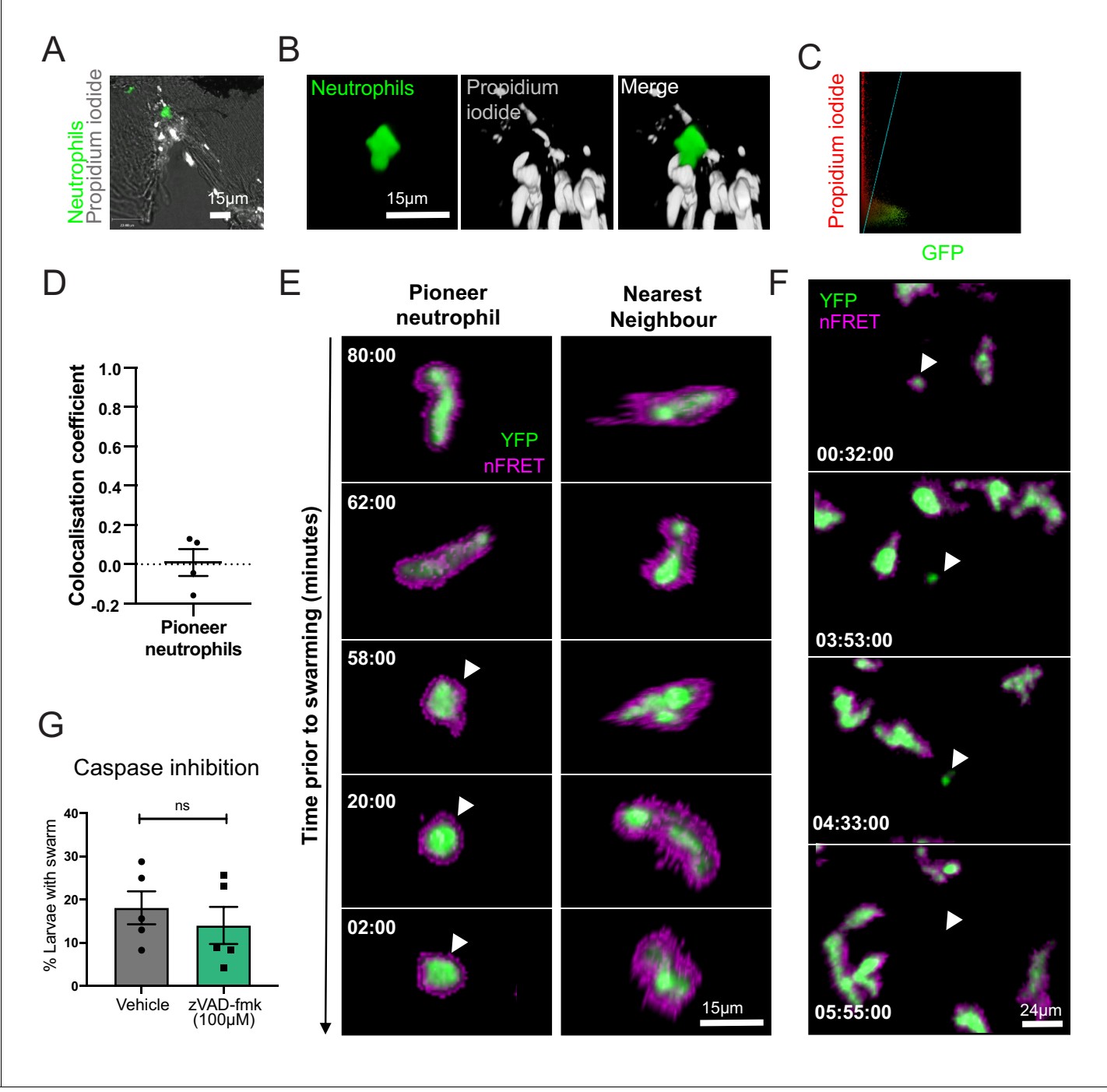

**Figure 3.** Pioneer neutrophils are not undergoing apoptosis prior to swarming. (**A–D**) Pioneer neutrophils are not propidium iodide positive prior to swarming. (**A**) Single slice image showing tail fin of injured *mpx:GFP* larva (bright field), stained with propidium iodide (white). Image shows representative example of pioneer neutrophil at the wound site prior to the swarming response (green). (**B**) Representative 3D render of pioneer neutrophil during the initiation phase. Left tile shows *mpx:GFP* pioneer neutrophil, middle tile shows propidium iodide staining, and right tile shows the two merged. (**C**) Representative colocalisation analysis of pioneer neutrophils, where neutrophil signal (GFP) is on the x axis and propidium iodide signal (mCherry) is on the y axis. (**D**) Pearson's colocalisation coefficient for pioneer neutrophils (data shown are mean ± SEM, n = 4 larvae) (*Figure 3— source data 1*). (**E**) Pioneer neutrophils are not apoptotic prior to swarming. 3 dpf *mpx:FRET* larvae were injured, and time lapse imaging was performed from 30 min post-injury for 6 hr. Neutrophil signal from the acceptor (green) and nFRET (magenta) are shown to illustrate neutrophil apoptosis. Representative example of a pioneer neutrophil and its nearest neighbour in the frames preceding neutrophil swarming. The initiation stage is observed 58 min prior to swarming (rounded pioneer neutrophil, arrowhead). nFRET signal is intact at all stages of migration prior to swarming in both the pioneer and nearest-neighbour non-pioneer neutrophil. Time stamps are mm:ss relative to the swarm start time (representative example of

*Figure 3 continued on next page*

Figure 3 continued

n = 6 neutrophils from five larvae). (**F**) Apoptotic neutrophils do not initiate swarming. Example of neutrophil apoptosis at the wound site demonstrated by loss of FRET signal around 4 hr post-injury (arrowhead), followed by the absence of neutrophil cluster formation in the same tissue region by the end of the imaging period. Time stamp is relative to injury time and is hh:mm:ss. (**G**) The percentage of larvae with neutrophil swarms at 3 hr post-injury after Caspase inhibition by zVAD-fmk or vehicle control treatment. Data shown are from n = 228 larvae accumulated from five biological replicates (*Figure 3—source data 2*). Paired data shows each individual experiment using the same batch of larvae over the treatment groups.

The online version of this article includes the following source data for figure 3:

**Source data 1.** Numerical data for the graph of *Figure 3D*.
**Source data 2.** Numerical data for the graph of *Figure 2G*.

---

supplement 1A,B). We analysed swarming neutrophils in *H2A-mCherry* larvae and identified that mCherry-positive material was released by swarming neutrophils in the cytoplasmic balloon structures (*Figure 6D,E*). In a manner analogous to mammalian NET release, the histone material extruded from the nucleus and was released in a catapult-like manner (*Figure 6D,E*). Neutrophil chromatin release was further confirmed using a second transgenic line *Tg(mpx:H2Bcerulean-P2A-mKO2CAAX)gl29* (*Manley et al., 2020*; *Figure 6—figure supplement 1C*). Neutrophil H2B histones colocalised with SYTOX green and myeloperoxidase (another well-defined NET component in mammalian systems) in vivo following calcium ionophore stimulation (*Figure 6—figure supplement 1D*). Together, these data demonstrate that zebrafish neutrophils exhibit co-ordinated release of NET components, including histones and myeloperoxidase, in vivo and that NET release occurs in developing neutrophil swarms.

## Inhibition of mediators of NET release decrease swarm frequency

Having identified that NETs are expelled by early swarming neutrophils, we next investigated mechanistic links between NET release and neutrophil swarming. DNA release in NET formation is mediated by a series of molecular events that lead to chromosome decondensation and the creation of pores in the nuclear/plasma membranes to allow release of nuclear material including chromatin. We sought to pharmacologically or genetically perturbate mediators of NET formation to assess their roles in swarming.

We first targeted gasdermin proteins that form pores in nuclear and plasma membranes to allow lytic release of intracellular components during NET formation and other forms of lytic cell death (e.g. pyroptosis) (*Chen et al., 2018*; *Sollberger et al., 2018*; *Chen et al., 2021*). To test the role of DNA release through gasdermin pores, larvae were pre-treated with the pore blocking gasdermin inhibitor LDC7559 prior to tissue injury. Treatment with LDC7559 reduced the percentage of larvae with swarms compared to DMSO control treatment (*Figure 7A*), suggesting that gasdermin pore formation is an important step in the swarming process.

Neutrophil elastase is known to play a key role in chromatin decondensation and NET release, by cleaving histone proteins allowing their release during NET formation (*Papayannopoulos et al., 2010*). Neutrophil elastase has also been reported to also cleave gasdermin D to facilitate pore formation (*Sollberger et al., 2018*). We used the cell-permeable inhibitor of neutrophil elastase MeOSu-AAPV-CMK (*Saffarzadeh et al., 2012*) to inhibit NET release downstream of NE. Systemic injection of MeOSu-AAPV-CMK resulted in a reduction in the percentage of larvae with neutrophil swarms at the injury site compared to DMSO controls (*Figure 7B*). These data further suggest a role for NET downstream of neutrophil elastase

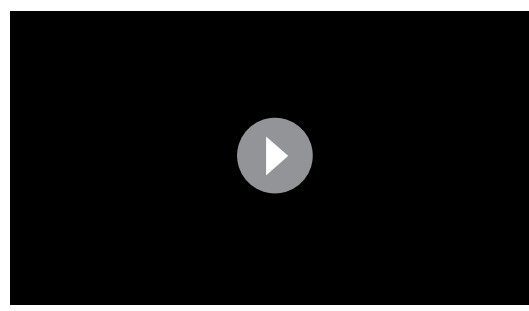

**Video 5.** Pioneer neutrophils are not undergoing apoptosis prior to swarming. The pioneer neutrophil (*mpx:GFP* positive) is propidium iodide (PI, white) negative, shown using a 3D confocal micrograph. The pioneer neutrophil (asterisk) is Caspase FRET positive in the Tg(mpx:CFP-DEVD-YFP)sh237 transgenic line shown by spinning disk confocal timelapse with FRET. https://elifesciences.org/articles/68755#video5

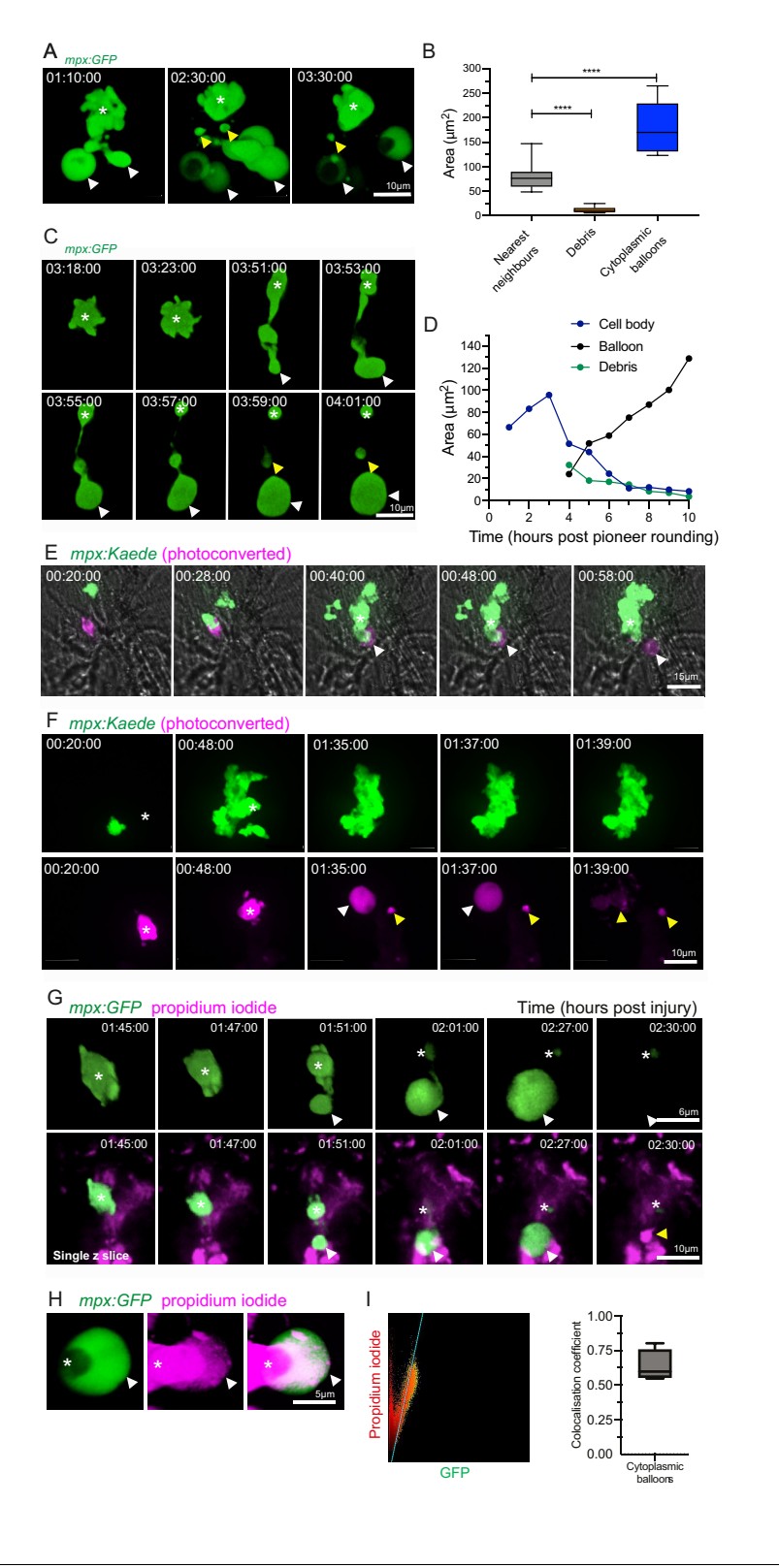

**Figure 4.** Catapult release of DNA-containing, balloon-like structures by pioneer neutrophils. (**A**) 3D rendered time lapse sequence showing cell fragments around swarming neutrophils (swarm: asterisk, cytoplasmic balloon-like structures: white arrowheads, cell debris: yellow arrowheads). Time stamps are hh:mm:ss relative to injury time. Images were taken using a 40× objective lens on a Perkin-Elmer spinning disk confocal microscope. (**B**) The area of cellular debris and cytoplasmic vacuoles detected during swarm aggregation were measured alongside three nearest neighbour neutrophils at

*Figure 4 continued on next page*

*Figure 4 continued*

the wound site (error bars are SEM. Groups were analysed using an ordinary one-way ANOVA with Tukey's multiple comparison, p<0.0001. N = 5 larvae) (*Figure 4—source data 1*). (C, D) Timelapse microscopy of catapult cytoplasmic balloon (white arrowhead) release from a single pioneer neutrophil (asterisk), leaving cell debris (yellow arrowhead). A plot of the areas of this event demonstrates that the cytoplasmic balloon grows as the cell body decreases (*Figure 4—source data 2*). The fluorescence of the debris is lost over the course of the timelapse. (E) A photoconversion approach to study pioneer neutrophils within developing clusters. 3 dpf *mpx:Kaede* larvae were injured and the neutrophil closest to the wound site was photoconverted from green to red fluorescence at 10 min post-injury. Larvae where the red neutrophil became the swarm-initiating pioneer neutrophil were analysed. Example time lapse of green wild-type swarming neutrophils which cluster around the magenta pioneer neutrophil. A swarm forms around this pioneer (asterisk) while a cytoplasmic balloon is released from the pioneer between 48–58 min post injury (identifiable from the magenta, white arrowhead). Time stamps are hh:mm:ss relative to time post injury. (F) A second example of a photoconverted pioneer neutrophil (asterisk) releasing an extracellular balloon (white arrowhead) and leaving cell debris in and around the swarm (yellow arrowheads). (G) A single confocal Z slice of propidium iodide staining (magenta) demonstrates that a pioneer neutrophil (asterisk), is PI negative until cytoplasmic balloon release (white arrowhead) that becomes PI positive over time (yellow arrowhead) and loses its green fluorescence. (H) A neutrophil (asterisk) cytoplasmic balloon (white arrowhead) becoming positive for propidium iodide. (I) Colocalisation of propidium iodide with neutrophil cytoplasmic fragments (*Figure 4—source data 3*).

The online version of this article includes the following source data for figure 4:

**Source data 1.** Numerical data for the graph of *Figure 4B*.
**Source data 2.** Numerical data for the graph of *Figure 4D*.
**Source data 3.** Numerical data for the graph of *Figure 4I*.

---

activity in modulating the zebrafish swarming response.

Neutrophil-derived antimicrobial mechanisms, such as NADPH oxidase and myeloperoxidase (MPO), have been implicated in neutrophil swarming in vitro (*Hopke et al., 2020*). Furthermore, MPO is a component of NETs, and its enzymatic activity has been implicated in NET formation, especially in studies involving stimulation by PMA, and can work synergistically with neutrophil elastase to further aid chromatin decondensation (*Papayannopoulos et al., 2010*). The NADPH-oxidase inhibitor diphenyleneiodonium chloride (DPI) did not reduce the percentage of larvae with swarms; however, DPI greatly reduced neutrophil recruitment to the wound if pre-treated at 4 hr pre-wound (*Figure 7—figure supplement 1A*); therefore, DPI was administered at 1 hpw to allow initial neutrophil recruitment to occur before DPI's effect on swarming events could be assessed (*Figure 7—figure supplement 1B*). We used CRISPR-Cas9 technology to target and knockdown the *myeloid-specific peroxidase* (*mpx*) promoter in zebrafish. We injected double transgenic *mpx:GFP; lyz:nfsB-mcherry* larvae with a crRNA targeting the *mpx* promoter. Knockdown was confirmed by a reduction in *mpx:GFP* expression compared to *tyr* CRISPant controls (*Figure 7C*), while *lyz:nfsB-mcherry* expression remained unchanged. Neutrophil swarms were subsequently assessed at 4 hpi by *lyz:nfsB-mcherry* expression. The percentage of larvae with swarms was reduced in *mpx* knockdown larvae when compared to *tyr* controls (*Figure 7D*), indicating a role for mpx in neutrophil swarming.

Together, these data demonstrate that targeting NET formation using independent approaches caused a reduction in neutrophil swarming, highlighting important roles for Gasdermin D, neutrophil elastase, and myeloid-specific peroxidase-dependent NET release in modulating the in vivo swarming response.

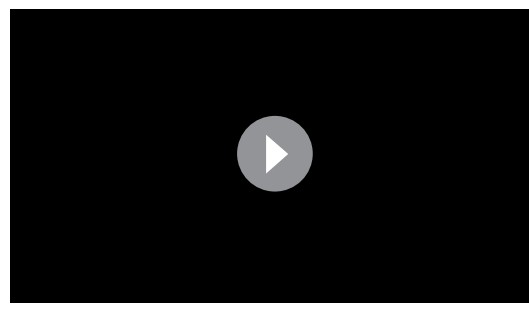

**Video 6.** Catapult release of cytoplasmic balloon-like structures by pioneer neutrophils. A spinning disk confocal timelapse of a nick wound in the *mpx:GFP* transgenic shows balloon-like structures (white arrowheads) released from the developing swarm. In the *mpx:Kaede* transgenic line the pioneer neutrophil was photoconverted to red and red cytoplasmic balloons were released at the wound (white arrowheads). Cytoplasmic balloons released from the swarming neutrophils become positive for propidium iodide (PI, red) indicated by white arrowheads.
https://elifesciences.org/articles/68755#video6

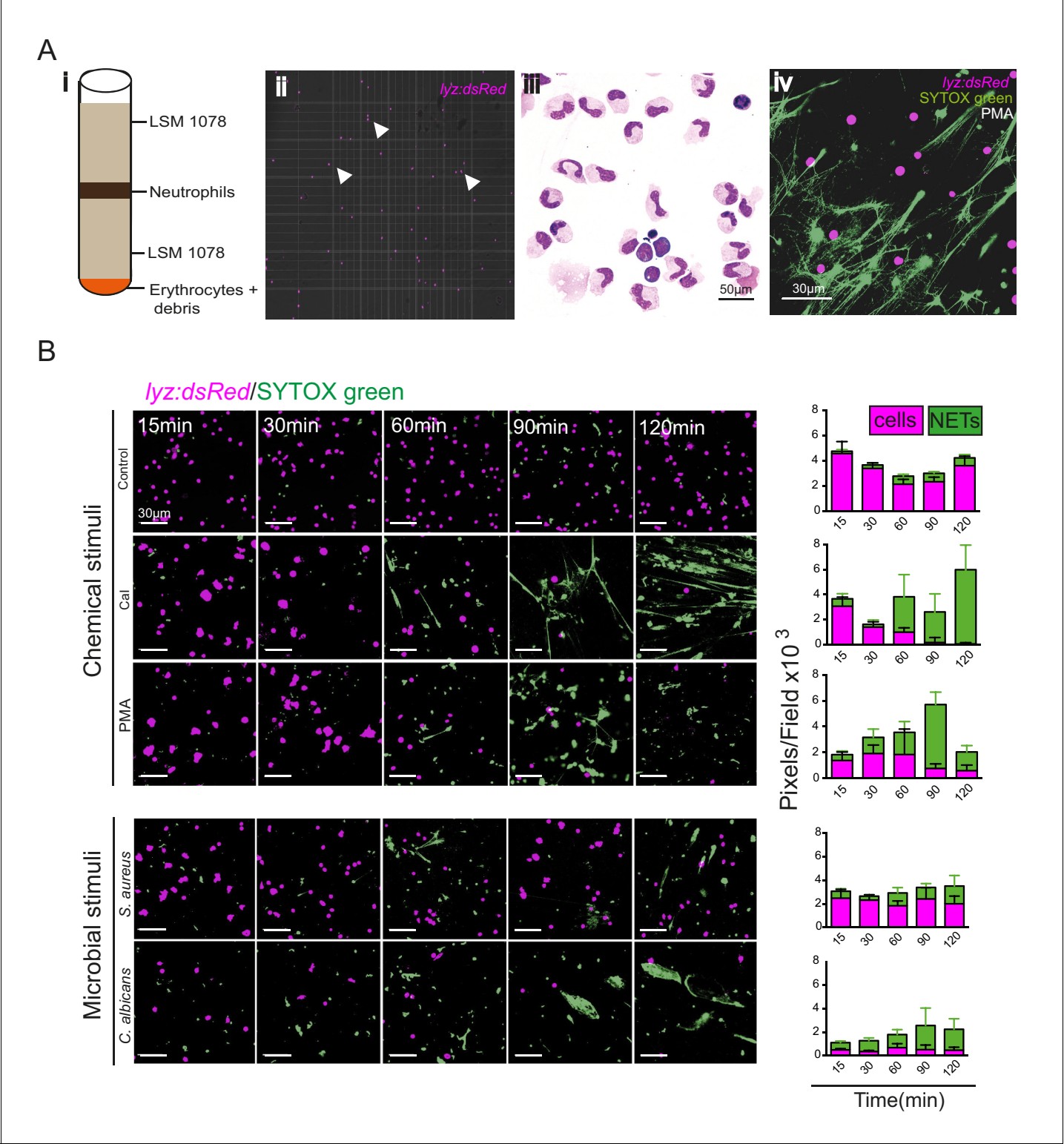

**Figure 5.** Zebrafish neutrophils release NETs in vitro. (**A**) Purification of adult zebrafish neutrophils for in vitro stimulation. Viable neutrophils were purified from the kidneys of adult *Tg(lyz:dsRed)nz50* adult zebrafish and separated by density gradient separation purification (i). Unstimulated lyz: dsRed neutrophils were visualised using a haemocytometer (ii) and by May Grünwald-staining of cytospin preparations (iii). NET release was observed using SYTOX green staining (example shown is after PMA treatment) (iv). (**B**) Morphological time course of NET release by *Tg(lyz:dsRed)nz50* following in vitro stimulation after no stimulation (control), calcium ionophore treatment, PMA treatment, *Staphylococcus aureus* infection, or *Candida albicans* infection. Images shown are randomly selected fields of view. Neutrophils were detected by their transgenic dsRED signal (magenta) and the

*Figure 5 continued on next page*

*Figure 5 continued*

extracellular DNA of NETs was detected using SYTOX (green). Graphs display total magenta and green pixel values. Data are mean ± SD for 10 random fields of view/timepoint.

## Discussion

In this study, we investigated neutrophil migration patterns in the context of inflammation and infection, demonstrating that neutrophil swarming behaviour is a part of zebrafish immunity. We focused on neutrophil swarming in injury-induced inflammation, where the zebrafish model allowed us to track endogenous neutrophils in an intact tissue damage model in vivo. Utilising the optical transparency of zebrafish larvae and a combination of transgenic reporter lines and fluorescent DNA intercalating agents, we identified that swarm-initiating, pioneer neutrophils release extracellular chromatin from within swarms, building on a growing body of work that implicates a role for pioneer neutrophil DNA release in the swarming response (*Lämmermann et al., 2013*; *Ng et al., 2011*; *Uderhardt et al., 2019*; *Byrd et al., 2013*). Finally, we block NET forming components and demonstrate a reduction in swarming events, drawing a mechanistic link between NET release and swarm formation. These findings build on in vitro evidence that swarming neutrophils release NETs (*Hopke et al., 2020*), and reveal a functional role for NET release in in vivo neutrophil swarming.

We used the visual transparency of the larval zebrafish model to precisely track neutrophils using fluorescence microscopy over time, providing high-resolution in vivo characterisation of endogenous neutrophil migration patterns in the context of swarming at sites of tissue injury. Within the inflamed tailfin, we show that neutrophil swarms developed around an individual pioneer neutrophil, sharing common behaviours with the pioneer neutrophils essential for swarm initiation in murine models (*Lämmermann et al., 2013*; *Uderhardt et al., 2019*; *Lämmermann, 2016*). Due to the relatively small number of neutrophils present in zebrafish larvae (~300), in comparison with the thousands (2–5 × $10^4$) (*Ng et al., 2011*) injected into the mouse ear, we could observe single-cell behaviours enabling us to study endogenous pioneer neutrophils with optical clarity prior to the onset of swarming. Within inflamed or infected interstitial tissue, the initial arrest of a small number of 'pioneer' or 'scouting' neutrophils precedes a later influx of neutrophil migration (*Lämmermann et al., 2013*; *Chtanova et al., 2008*; *Uderhardt et al., 2019*). We distinguish the pioneer neutrophil from other scouting neutrophilsand propose that the pioneer neutrophil release extracellular DNA that precedes consolidation of swarm formation through an aggregation phase.

Neutrophil responses to tissue injury in mammalian systems are bi-phasic and are modulated in part by the lipid LTB4, which acts as a signal-relay molecule to amplify initial signals produced at inflammatory sites including formyl peptides (*Afonso et al., 2012*; *Reátegui et al., 2017*; *Lämmermann et al., 2013*). We demonstrate that zebrafish neutrophil recruitment to tail fin inflammation is also bi-phasic and requires intact *lta4h* and *blt1*. In recent findings, LTB4 biosynthesis is increased by swarming neutrophils following a sustained calcium alarm signal during neutrophil swarming, further demonstrating a conserved role of LTB4 in zebrafish (*Poplimont et al., 2020*).

We observed that pioneer neutrophils are surrounded by debris and extracellular DNA. This is consistent with the recent identification that early swarming neutrophils sense DAMPs, including ATP released by necrotic tissue, which causes a sustained calcium wave, translocation of 5-LO, and subsequent metabolism of arachidonic acid to LTB4 (*Poplimont et al., 2020*). We demonstrate that a caspase-3-sensitive FRET reporter fluorescence was intact during the swarm initiation phase and caspase inhibition had no effect on swarming outcomes, suggesting that swarm initiating pioneer neutrophils were not undergoing neutrophil apoptosis prior to swarming, despite adopting a characteristic rounded morphology. Here we show that pioneer neutrophils are propidium iodide negative, suggesting that the plasma membrane remains intact, something that has been difficult to conclusively show in other models (*Lämmermann et al., 2013*; *Uderhardt et al., 2019*). Interestingly, the distinct release of large balloon-like structures, which become positive for propidium iodide, have been recently identified in zebrafish models of laser-induced tissue injury (*Poplimont et al., 2020*), as well as mechanical tissue injury (*Manley et al., 2020*). Our data build on this observation, identifying that extracellular DNA is released from pioneer and scouting neutrophils during early swarming, in a catapult-like fashion reminiscent of mammalian NET release.

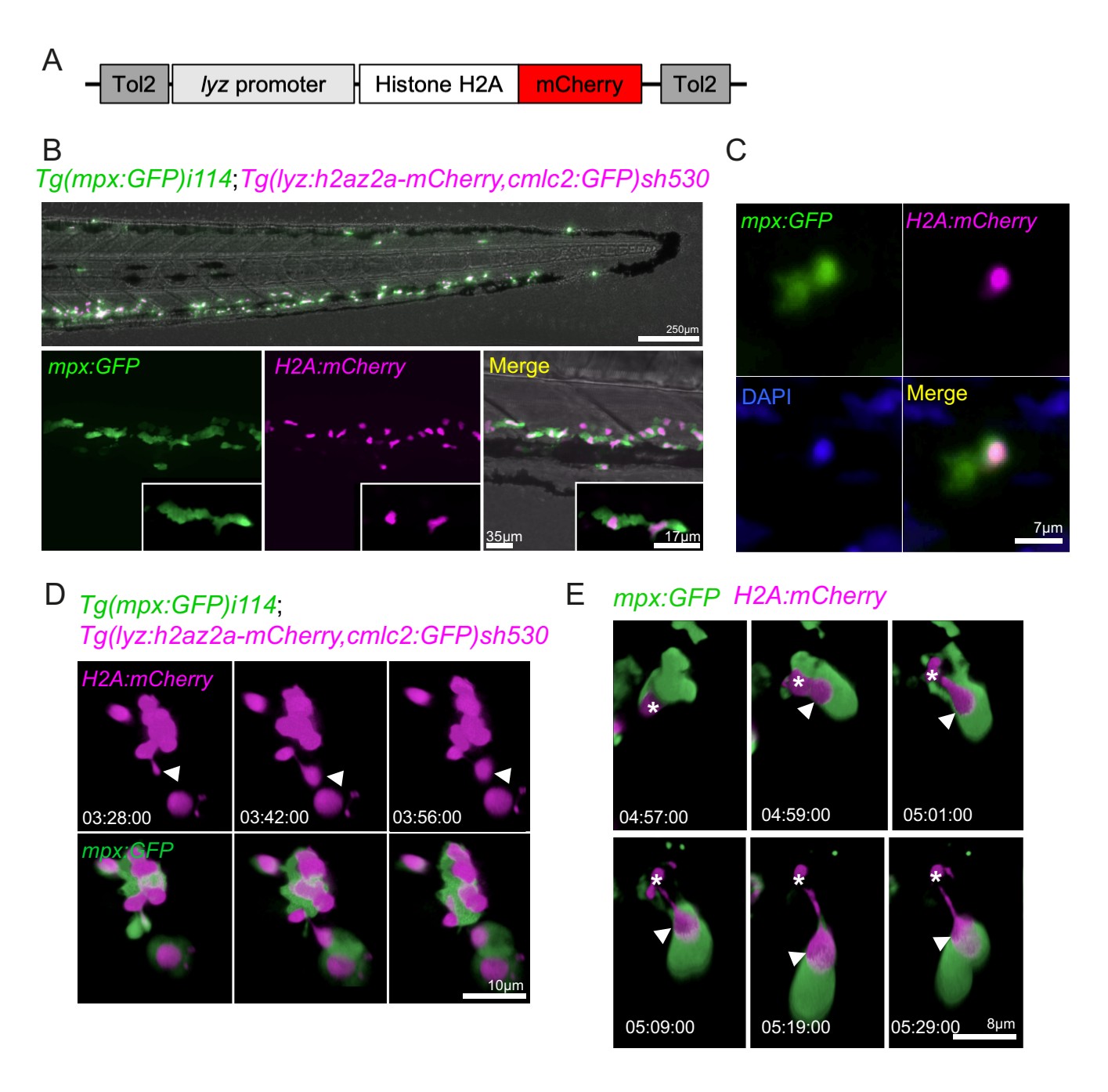

**Figure 6.** Neutrophils release histones early in swarm formation. (**A**) Schematic of the *lyz:H2A.mCherry* construct made by Gateway cloning, which includes the neutrophil specific promoter (lyz), and the histone H2A gene fused to the fluorescent protein mCherry flanked by Tol2 arms to aid transgenesis. (**B, C**) Representative image of the stable *TgBAC(mpx:GFP)i114;Tg(lyz:H2A.mCherry)sh530* transgenic line. (**B**) Image shows the caudal haematopoietic tissue of a 3 dpf sh530 larvae, where the H2A mCherry transgene is expressed in neutrophils. (**C**) 40× confocal image of the transgenic line, showing neutrophil histones labelled by the transgene. (**D**) Representative example of NET release from swarming neutrophils from six larvae. Time course of *Tg(mpx:GFP)i114;Tg(lyz:h2az2a-mCherry,cmlc2:GFP)sh530* larva showing a single neutrophil and histone H2A (white arrows), undergoing NET-like morphological changes where histones are released from the centre of swarms in cytoplasmic vesicles. Time stamps are hh:mm:ss relative to time post injury. (**E**) A second example timelapse showing that histones are extruded from the nucleus during the cytoplasmic balloon release.

The online version of this article includes the following source data and figure supplement(s) for figure 6:

**Figure supplement 1.** Validation of zebrafish histone transgenic lines.

**Figure supplement 1—source data 1.** Numerical data for the graph of *Figure 6—figure supplement 1A*.

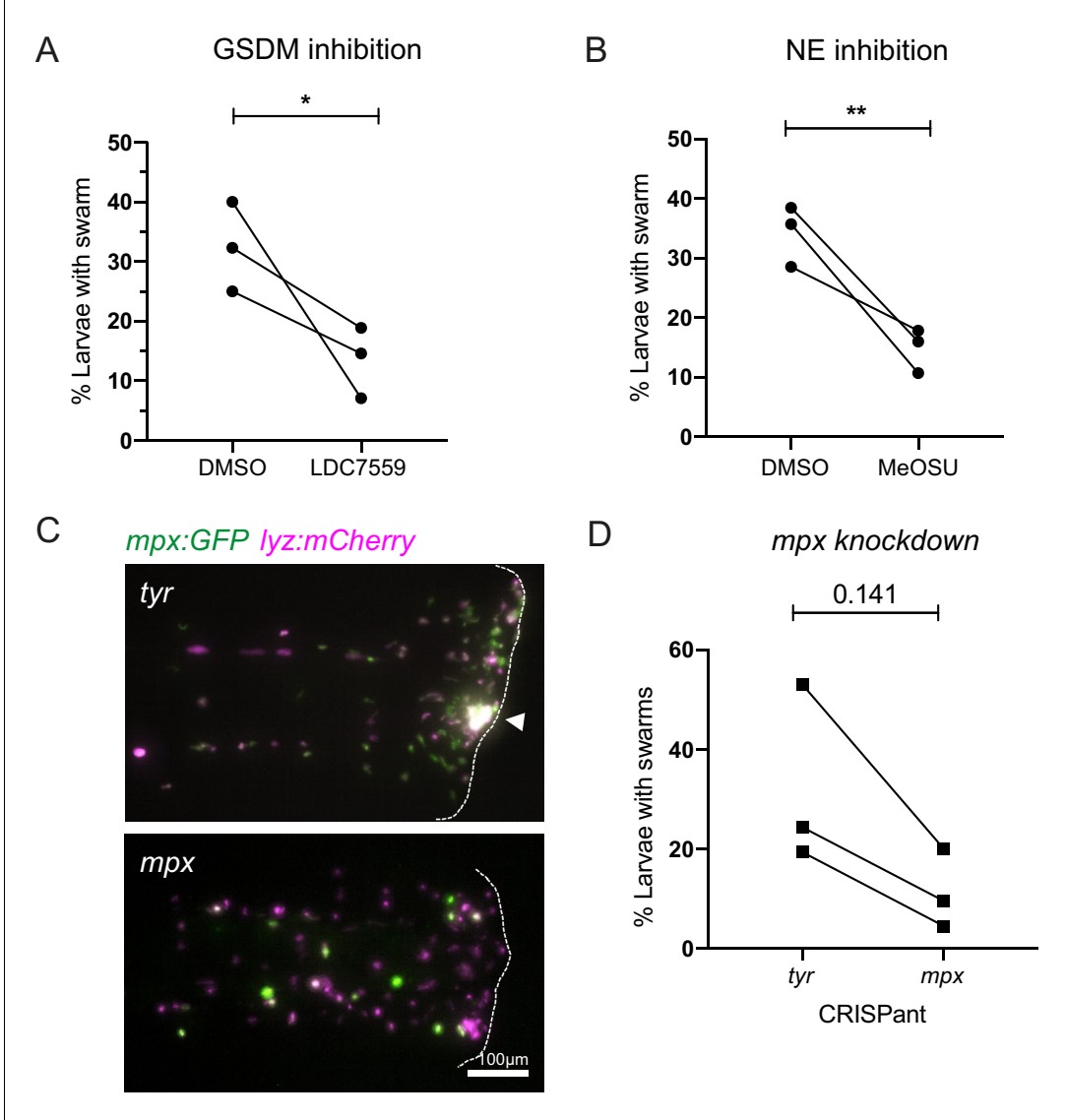

**Figure 7.** Inhibition of NET components, gasdermin D, neutrophil elastase, and myeloperoxidase decrease swarm frequency. (A) The percentage of larvae with neutrophil swarms at 3 hr post injury after gasdermin inhibitor, LDC7559, or DMSO treatment. Data shown are mean with a minimum of 120 larvae analysed in each group, accumulated from three biological replicates. Joined data represent each individual experiment using larvae from the same zebrafish lay over the two treatment groups (*Figure 7—source data 1*). (B) The percentage of larvae with neutrophil swarms at 4 hr post injury after neutrophil elastase inhibitor, MeOSu-AAPV-CMK, treatment, or DMSO control. Data shown are mean, with greater than 73 larvae per group over three biological replicates. Joined data represent each individual experiment using larvae from the same zebrafish lay over the two treatment groups (*Figure 7—source data 2*). (C) Representative fluorescence micrographs of the double transgenic *Tg(mpx:GFP);Tg(lyz:nfsβ-mCherry)* after *myeloperoxidase* knockdown using CRISPR-Cas9 with *tyrosinase* knockdown as a negative control. The *myeloperoxidase* guide RNA targeted the promoter of *mpx*, therefore knocking down expression of green *mpx:GFP* while leaving *lyz:mCherry* intact. White arrowhead indicates the presence of a swarm at the wound (white dashed line). (D) The percentage of larvae with neutrophil swarms at 4 hr post-injury after *mpx* knockdown by CRISPR-Cas9 or *tyr* control. Data shown are mean, with a minimum of 115 larvae analysed in each group, accumulated from three biological replicates (*Figure 7—source data 3*). Joined data represent each individual experiment using larvae from the same zebrafish lay over the two treatment groups. p-values in (A), (C), and (D) are generated from unpaired t-tests.

The online version of this article includes the following source data and figure supplement(s) for figure 7:

**Source data 1.** Numerical data for the graph of *Figure 7A*.
**Source data 2.** Numerical data for the graph of *Figure 7B*.
**Source data 3.** Numerical data for the graph of *Figure 7D*.
**Figure supplement 1.** ROS inhibition by DPI and swarm frequency.

Here we show that zebrafish neutrophils can release NET components using mammalian NET-inducing stimuli PMA and calcium ionophore, as well as more physiologically relevant microbial infection. We confirm that zebrafish NETs contained chromatin and myeloperoxidase, providing some of the first evidence that zebrafish neutrophils can release NETs and contain multiple components observed in mammalian systems (*Brinkmann et al., 2004*; *Chen et al., 2021*).

Whilst imaging of NETs in vivo is advancing, the understanding of the kinetics of DNA release from neutrophils remains limited. Here, we have developed in vivo zebrafish reporter transgenic lines for live imaging of chromatin release using fluorescent protein-tagged histones. Imaging data from other groups corroborate our identification that NET releasing neutrophils undergo distinct morphological changes involving the stretching of neutrophil cytoplasm and production of large extracellular DNA containing vesicles and cellular debris (*Tanaka et al., 2014*; *Pilsczek et al., 2010*). Following LPS stimulation, murine neutrophils expel extracellular DNA in large, balloon-like cytoplasmic vesicles, similar to our observations in zebrafish (*Tanaka et al., 2014*). In human neutrophils infected with *S aureus*, vesicles containing DNA are released into the extracellular space where they lyse and release their contents to form NETs, consistent with our observation that cytoplasmic balloon-like structures containing extracellular DNA are released by neutrophils (*Pilsczek et al., 2010*). The catapult-like release of DNA and histones by neutrophils observed in our experiments is consistent with in vitro evidence that DNA release by eosinophils is catapult-like in its expulsion (*Yousefi et al., 2008*). We show that zebrafish NETs contain granular proteins consistent with those observed in mammalian neutrophils, adding to the evidence that NET release is found in zebrafish in vivo (*Johnson et al., 2018*; *Palić et al., 2007*). Our experiments enable the morphology of NET-releasing neutrophils to be observed in vivo, in real time, providing new insight into the morphological changes associated with NET release.

We set out to determine the mechanism of NET release in relation to swarming. The gold standard for NET depletion in vitro is the addition of exogenous DNase to degrade NET DNA. Administration of a range of DNase1 concentrations by incubation in the water of larvae was highly toxic to live zebrafish (data not shown); hence, we were not able to assess the role of extracellular DNA in swarm formation in this way. We therefore targeted components of the NETosis pathway to investigate the relationship between NET release and swarming. NET release is mediated by MPO activation, and release of NE into the cytoplasm. This leads to chromatin decondensation and nuclear membrane breakdown, resulting in the mixing of nuclear chromatin with granular and cytoplasmic proteins. DNA release in NET formation (*Chen et al., 2018*; *Sollberger et al., 2018*), and other forms of lytic release (e.g. pyroptosis; *Chen et al., 2021*), requires gasdermin proteins that forms pores in the nuclear and plasma membranes to allow lytic release of intracellular components. Humans possess six isoforms (GSDMA, GSDMB, GSDMC, GSDMD, GSDME, and PJVK) and in zebrafish *pjvk*, *gsdmea*, and *gsdmeb* have been identified as orthologues. A recent zebrafish study identified that activation of neutrophil pyroptosis is essential for NET formation during hemolysin-overexpressing *E. piscicida* (EthA$^+$) *E. piscicida* infection and that this is mediated by the caspy2-GSDMEb axis (*Chen et al., 2021*). By inhibiting pore-forming gasdermins, we found that the swarming response was reduced linking gasdermin-mediated chromatin release to neutrophil swarming in vivo. Neutrophil elastase-deficient mice have defective NET formation, although some reports suggest that they can still be made in response to strong stimuli such as PMA (*Martinod et al., 2016*). Humans have a large repertoire of elastase proteins (CELA1, CELA2A, CELA2B, CELA3A, CELA3B, ELANE, AZU1, CTRC, and PRTN3. CELA1 to CELA3B), of which PRTN3, ELANE, and AZU1 are in neutrophils (*Wright et al., 2013*). Zebrafish have orthologues of *cela*, *ela*, and *prtn3* genes, but have multiple copies of each annotated on the genome (*cela1.1*, *cela1.2*, *cela1.3*, *cela1.4*, *cela1.5*, *cela1.6*, *ela2*, and *ela2l*), rendering genetic approaches unfeasible. By targeting NE pharmacologically, we identified a reduction in neutrophil swarming in zebrafish, further supporting a functional role of NE in the swarming response. These findings contribute important in vivo data to growing evidence that NETs contribute to neutrophil swarming. NET release is observed at sites of alum injection-associated with neutrophil swarming in mice (*Stephen et al., 2017*), and NET formation facilitates neutrophil aggregation at sites of fungal infection (*Byrd et al., 2013*). Neutrophil swarming in zebrafish has recently been shown to be mediated by a sustained calcium alarm signal following sensing of danger signals including ATP, which are propagated by neutrophils via Cx43 connexins (*Poplimont et al., 2020*). Deciphering the intricate modulation of NET release, danger signal release, and calcium signalling in swarm formation requires further mechanistic studies in complementary models.

Interestingly, inhibition of caspases did not appear to contribute to swarm frequency in our zebrafish studies. Our images and videos strongly suggest a form of lytic cell death of pioneer neutrophils is occurring, leading to cell debris and death of the cell body, not consistent with some reports of vital NET formation in other models (*Tong et al., 2019*; *Lelliott et al., 2020*; *Desai et al., 2016*). However, the mode of this cell death remains undefined and challenging to address in vivo due to lack of available tools to define this. In mammalian systems an important, but not critical, player in the onset of NET forming pathways is NADPH-oxidase-induced reactive oxygen species (ROS) release (*Papayannopoulos et al., 2010*; *Fuchs et al., 2007*). ROS inhibition via DPI did not decrease swarm frequency in the zebrafish model. This is consistent with recent findings that DPI does not inhibit the formation of neutrophil swarms in synchronised in vitro swarm arrays. However, DPI did decrease swarm stability around foci of fungal infections in swarm arrays (*Hopke et al., 2020*); therefore, understanding the roles of NADPH-oxidase in the different phases of swarming requires further investigation in vivo.

Our findings in this study implicate a role for endogenous pioneer neutrophil NET release in swarming. Our zebrafish data demonstrate that it is possible to dissect mechanisms of NET and swarm formation in vivo, mechanisms that have been challenging to uncover in endogenous neutrophils in mammalian models. Understanding why swarms are initiated is important for understanding the signals that control the coordination of neutrophil migration within interstitial tissue, which ultimately could lead to the identification of novel therapeutic avenues to target excessive inflammation for the treatment of chronic inflammatory disease.

# Materials and methods

## Key resources table

| Reagent type (species) or resource | Designation | Source or reference | Identifiers | Additional information |
|---|---|---|---|---|
| Gene (*Danio rerio*) | TgBAC (mpx:EGFP)i114 | Renshaw et al., Blood 2011 | i114Tg RRID:ZFIN_ZDB-ALT-070118-2 | Transgenic |
| Gene (*Danio rerio*) | Tg(lyz:nfsβ-mCherry)sh260 | *Buchan et al., 2019* | sh260Tg RRID:ZFIN_ZDB-ALT-190925–14 | Transgenic |
| Gene (*Danio rerio*) | TgBAC(mpx: GAL4-VP16)sh256 | Prajsnar et al., Infection and Immunity, 2013 | sh256Tg RRID:ZFIN_ZDB-ALT-131203–1 | Transgenic |
| Gene (*Danio rerio*) | Tg(UAS:kaede)s1999t | *Isles et al., 2019a* | s1999tTg RRID:ZFIN_ZDB-ALT-070314–1 | Transgenic |
| Gene (*Danio rerio*) | TgBAC(mpx:CFP-DEVD-YFP)sh237 | *Robertson et al., 2016* | sh237Tg RRID:ZFIN_ZDB-ALT-161012–5 | Transgenic |
| Gene (*Danio rerio*) | Tg(lyz:h2az2a-mCherry,cmlc2:GFP)sh530 | This paper | sh530Tg | Transgenic |
| Gene (*Danio rerio*) | Tg(mpx:H2Bcerulean-P2A-mKO2CAAX)gl29 | Manley et al., Journal of Leukocyte Biology and This paper | gl29Tg RRID:ZFIN_ZDB-ALT-151201–2 | Transgenic |
| Gene (*Danio rerio*) | Tg(lyz:dsRed)nz50 | Hall et al., BMC Developmental Biology 2019 | nz50Tg RRID:ZFIN_ZDB-ALT-071109–3 | Transgenic |
| Genetic reagent (*Danio rerio*) | ltah4 CRISPR-Cas9 guide RNA | This paper | *ltah4 CRISPR* | AGGGTCTGAAACTGGAGTCA(TGG) |
| Genetic reagent (*Danio rerio*) | blt1 CRISPR-Cas9 guide RNA | This paper | blt1 CRISPR | CAATGCCAATCTGATGGGAC(AGG) |
| Genetic reagent (*Danio rerio*) | myeloperoxidase CRISPR-Cas9 guide RNA | This paper | *mpx CRISPR* | GTTGTGCTGAATGTATGCAG(CGG) |
| Genetic reagent (*Danio rerio*) | tyrosinase CRISPR-Cas9 guide RNA | Isles et al., Frontiers in Immunology, 2019 | *tyr CRISPR* | GGACTGGAGGACTTCTGGGG(AGG) |
| Sequence-based reagent | lta4h_fw | This paper | PCR primer | GTGTAGGTTAAAATCCATTCGCA |

*Continued on next page*

*Continued*

| Reagent type (species) or resource | Designation | Source or reference | Identifiers | Additional information |
|---|---|---|---|---|
| Sequence-based reagent | *lta4h_rev* | This paper | PCR primer | GAGAGCGAGGAGAAGGAGCT |
| Sequence-based reagent | *blt1_fw* | This paper | PCR primer | GTCTTCTCTGGACCACCTGC |
| Sequence-based reagent | *blt1_rev* | This paper | PCR primer | ACACAAAAGCGATAACCAGGA |
| Recombinant DNA reagent | p5E-MCS lyz | *Kwan et al., 2007* | Plasmid | Gateway compatible plasmid |
| Recombinant DNA reagent | p3E-PolyA | *Kwan et al., 2007* | Plasmid | Gateway compatible plasmid |
| Recombinant DNA reagent | pDestTol2CG2 | *Kwan et al., 2007* | Plasmid | Gateway destination vector |
| Recombinant DNA reagent | pME-h2a-mCherry | This paper | Plasmid | Gateway compatible plasmid |
| Strain, strain background (*Staphylococcus aureus*) | SH1000 pMV158mCherry | *Pollitt et al., 2018* | SH1000 | Transgenic |
| Antibody | anti-mpx (rabbit polyclonal) | GeneTex | GeneTex: GTX128379 | (1:200) |
| Antibody | anti-eGFP (chicken polyclonal) | Abcam | Abcam: ab13970 | (1:2000) |
| Antibody | anti-rabbit Alexafluor 647 (goat polyclonal) | Jackson ImmunoResearch | Jackson Immuno Research: 111-605-045 | (1:1000) |
| Antibody | anti-chicken Alexafluor 488 (goat polyclonal) | Jackson ImmunoResearch | Jackson Immuno Research: 103-545-155 | (1:1000) |
| Chemical compound, drug | zVAD-fmk | Santa Cruz Biotechnology | Z-VAD-FMK (CAS 187389-52-2) | |
| Chemical compound, drug | LDC7559 | MedChem Express | CAS No.: 2407782-01-6 | |
| Chemical compound, drug | MeOSu-AAPV-CMK | Sigma-Aldrich | CAS No.: 65144-34-5 | |
| Chemical compound, drug | Diphenyleneiodonium chloride (DPI) | Sigma-Aldrich | CAS No.: 4673-26-1 | |
| Software, algorithm | NIS elements | Nikon | https://www.microscope.healthcare.nikon.com/products/software/nis-elements | |
| Software, algorithm | MatLab | MathWorks | https://www.mathworks.com/products/matlab.html | |
| Software, algorithm | BASiCz | Blood atlas of single cells in zebrafish | https://www.sanger.ac.uk/tool/basicz/ | |
| Software, algorithm | ChopChop | ChopChop | http://chopchop.cbu.uib.no/ | |
| Software, algorithm | Primer3 | ELIXIR | https://primer3.ut.ee/ | |
| Software, algorithm | Volocity | Quorum Technologies | https://quorumtechnologies.com/volocity | |
| Other | lta4h Synthetic SynRNA | Merck | lta4h CRISPR guide RNA | AGGGTCTGAAACTGGAGTCA(TGG) |
| Other | blt1 Synthetic SynRNA | Merck | blt1 CRISPR guide RNA | CAATGCCAATCTGATGGGAC(AGG) |
| Other | mpx Synthetic SynRNA | Merck | mpx CRISPR guide RNA | GTTGTGCTGAATGTATGCAG(CGG) |
| Other | tyr Synthetic SynRNA | Merck | tyr CRISPR guide RNA | GGACTGGAGGACTTCTGGGG(AGG). |
| Other | tracrSynthetic SynRNA | Merck | tracr CRISPR RNA | |

## Zebrafish husbandry and ethics

To study neutrophils during inflammation, *TgBAC(mpx:EGFP)i114*, *Tg(lyz:nfsβ-mCherry)sh260*, *TgBAC(mpx:GAL4-VP16)sh256Tg(UAS:kaede)s1999t*, *TgBAC(mpx:CFP-DEVD-YFP)sh23*, *TgBAC(mpx:GFP)i114;Tg(lyz:h2az2a-mCherry,cmlc2:GFP)sh530* and *Tg(mpx:H2Bcerulean-P2A-mKO2CAAX)gl29* zebrafish larvae were bred to produce larvae. All zebrafish were raised in the Biology Services Aquarium (BSA) at the University of Sheffield in UK Home Office-approved aquaria or in the Fish-Core aquarium at Monash University, Melbourne, Australia, and were maintained following standard protocols (*Nüsslein-Volhard and Dahm, 2002*). Adult fish were maintained at 28°C with a continuous re-circulating water supply and a daily light/dark cycle of 14/10 hr. All procedures were performed on larvae less than 5.2 dpf, which were therefore outside of the Animals (Scientific Procedures) Act, to standards set by the UK Home Office. Animal experiments performed in Australia conformed to 'Australian code for the care and use of animals for scientific purposes (2013)' and were undertaken under protocol MAS/2010/18 approved by the MARP2 Animal Ethics Committee at Monash University.

## Tail fin transection assay

To induce an inflammatory response, zebrafish larvae at 2 or 3 dpf were anaesthetised in 4% Tricaine (0.168 mg/ml; Sigma-Aldrich) in E3 media and visualised under a dissecting microscope. For linear tail fin injury, tail fins were transected consistently using a scalpel blade (5 mm depth, WPI) by slicing immediately posterior to the circulatory loop, ensuring the circulatory loop remained intact as previously described (*Renshaw et al., 2006*). For high-resolution imaging, tail fins were nicked by placing the tip of the scalpel blade directly below the end of the caudal vein and slicing through the ventral fin, such that the entire wound site could be observed using a 40× objective.

## Widefield microscopy of transgenic larvae

For neutrophil tracking experiments, injured 3 dpf *mpx:GFP* larvae were mounted in a 1% low-melting-point agarose solution (Sigma-Aldrich) containing 0.168 mg/ml tricaine immediately following tail fin transection. Agarose was covered with 500 µl of clear E3 solution containing 0.168 mg/ml tricaine to prevent dehydration. Time lapse imaging was performed from 0.5 to 5 hr post-injury with acquisition every 30 s using 10 z-planes were captured per larvae over a focal range of 100 µm using an Andor Zyla five camera (Nikon) and a GFP-specific filter with excitation at 488 nm. Maximum intensity projections were generated by NIS elements (Nikon) to visualise all 10 z-planes.

## Confocal microscopy of transgenic larvae

For visualising neutrophil swarming at high magnification, larvae were mounted in a 1% low melting point agarose solution (Sigma-Aldrich) containing 0.168 mg/ml tricaine for imaging immediately after tail fin transection. Agarose was covered with 2000 µl of clear E3 solution containing 0.168 mg/ml tricaine to prevent dehydration. Imaging was performed from 30 min post injury using a 20× or 40× objective on an UltraVIEWVoX spinning disk confocal laser imaging system (Perkin Elmer). Fluorescence for GFP was acquired using an excitation wavelength of 488 nm and emission was detected at 510 nm, fluorescence for DAPI was acquired using an excitation wavelength of 405 nm and emission was detected at 440 nm, and fluorescence for mCherry was acquired using 525 nm emission and detected at 640 nm. Images were processed using Volocity software.

## Tracking assays

Tracking of GFP-labelled neutrophils was performed using NIS Elements (Version 4.3) with an additional NIS elements tracking module. A binary layer was added to maximum intensity projections to detect objects. Objects were smoothed, cleaned, and separated to improve accuracy. A size restriction was applied where necessary to exclude small and large objects which did not correspond to individual neutrophils.

## Distance–time plots

For wound plots, the distances from the wound were obtained by processing neutrophil tracks under the assumption that the tail fin wound is a straight line parallel to the x-axis of the greyscale image. Neutrophil tracking data was extracted from NIS elements and imported into MatLab software. For

distance to pioneer plots, the pioneer centre was set as a reference point and tracking was performed to determine neutrophil distance to the reference point. Tracks were extracted from NIS elements and plotted manually using GraphPad Prism version 8.0.

## Neutrophil-specific expression of zebrafish genes

Gene expression was assessed using an RNA sequencing database from fluorescence-activated cell sorting (FACS) sorted GFP-positive cells from 5 dpf zebrafish (*Rougeot et al., 2019*) (data deposited on GEO under accession number GSE78954). RPKM values for genes of interest were extracted. For single-cell analysis, gene expression values were extracted from the BASiCz (Blood atlas of single cells in zebrafish) cloud repository (*Athanasiadis et al., 2017*). Cells of the neutrophil lineage were analysed for expression of LTB4 signalling components.

## CRISPR/Cas9 reagents

Synthetic SygRNA (crRNA and tracrRNA) (Merck) in combination with cas9 nuclease protein (Merck) was used for gene editing. Transactivating RNAs (tracrRNA) and gene-specific CRISPR RNAs (crRNA) were resuspended to a concentration of 20 µM in nuclease-free water containing 10 mM Tris–HCl pH 8. SygRNA complexes were assembled on ice immediately before use using a 1:1:1 ratio of crRNA:tracrRNA:Cas9 protein. Gene-specific crRNAs were designed using the online tool CHOP-CHOP (http://chopchop.cbu.uib.no/). We used the following crRNA sequences to target the ATG region of *blt1* and *lta4h*, where the PAM site is indicated in brackets: *lta4h:* AGGGTCTGAAAC TGGAGTCA(TGG), *blt1:* CAATGCCAATCTGATGGGAC(AGG). crRNA for *mpx* targets the promotor region, *mpx:* GTTGTGCTGAATGTATGCAG(CGG). Tyrosinase control crRNA *tyr:* GGACTGGAG-GACTTCTGGGG(AGG).

## Microinjection of SygRNA into embryos

A 1 nl drop of SygRNA:Cas9 protein complex was injected into *mpx:GFP* embryos or double transgenic *mpx:GFP;lyz:nfsB-mCherry* embryos at the one-cell stage. Embryos were collected at the one cell stage and injected using non-filament glass capillary needles (Kwik-Fil Borosilicate Glass Capillaries, World Precision Instruments (WPI), Herts, UK). RNA was prepared in sterile Eppendorf tubes. A graticule was used to measure 0.5 nl droplet sizes to allow for consistency of injections. Injections were performed under a dissecting microscope attached to a microinjection rig (WPI) and a final volume of 1 nl was injected.

## Genotyping and melting curve analysis

Site-specific mutations were detected using high-resolution melting (HRM) analysis, which can reliably detect CRISPR-Cas9-induced indels in embryos (*Samarut et al., 2016*; *Parant et al., 2009*). Genomic DNA extraction was performed on larvae at 2 dpf. Larvae were placed individually in 0.2 ml PCR tubes in 90 µl 50 mM NaOH and boiled at 95˚ for 20 min. Ten microlitre Tris–HCL ph8 was added as a reaction buffer and mixed thoroughly. Gene-specific primers were designed using the Primer three web tool (http://primer3.ut.ee/). Sequences were as follows: *lta4h_fw:* CGTGTAGG TTAAAATCCATTCGCA *lta4h_rev:* GAGAGCGAGGAGAAGGAGCT *blt1_fw:* GTCTTCTCTGGAC-CACCTGC *blt1_rev:* ACACAAAAGCGATAACCAGGA. HRM analysis (Bio-Rad) PCR were made with 5 µl Sybr Green master mix (Thermo Fisher), 0.5 µl of each primer (10 µM), 1 µl gDNA, and 3 µl water to make a final reaction volume of 10 µl. PCR were performed in a LightCycler instrument (Bio-Rad) using 96-well plates. The two-step reaction protocol was as follows: 95˚C for 2 min, followed by 35 cycles of 95˚C for 10 s, 58˚C for 30 s, 72˚C for 20 s. The second stage of the protocol was 95˚C for 30 s, 60˚C for 60 s, 65˚C for 10 s. The temperature then increased by 0.02˚C/s until 95˚C for 10 s. Melt curves were analysed using Bio-Rad software version 1.2. Successful detection of CRISPR-Cas9-induced indels is illustrated in *Figure 1—figure supplement 4B*. Mutagenesis frequencies of 91% and 88% were detected for *lta4h* and *blt1*, respectively.

## Preparation of microbial agents

*Staphylococcus aureus* strain SH1000 pMV158mCherry was used (*Pollitt et al., 2018*). An overnight bacterial culture was prepared by growing 1 cfu of SH1000 pMV158mCherry in 10 ml of bovine heart medium (BHI) (Sigma-Aldrich, lot number 53286) and 10 µl of 5 mg/ml tetracycline (Sigma-Aldrich)

for 16–18 hr at 37℃. Five hundred microlitre of this overnight culture was then aliquoted into 50 ml of BHI (Sigma Aldrich, 53286) infused with 50 microlitre of 5 mg/ml tetracycline (Sigma-Aldrich) and grown until an optical density at 600 nm of 0.5 was obtained. This culture was pelleted and resuspended in PBS (pH 7.4) (Fisher Scientific, lot number 1282 1680) to a concentration of 2500 cfu/nl. Phorbol myristate acetate (PMA, Sigma; stock 1 mg/ml) and calcium ionophore (CaI, Sigma; stock 1 mg/ml) were prepared in dimethyl sulfoxide (DMSO, Sigma) and stored at −70℃.

## Otic vesicle injections

*S. aureus:* 2500 cfu of Sh1000 pMV158mCherry was injected into the otic vesicle of 2 dpf *Tg(mpx: GFP)i114* larvae. Injections were performed under a dissecting microscope attached to a microinjection rig (WPI) and a final volume of 1 nl was injected. For analysis of swarm volumes, larvae were fixed in 4% paraformaldehyde in PBS and imaged using an UltraVIEWVoX spinning disk confocal laser imaging system (Perkin Elmer).

## FRET imaging of neutrophil apoptosis

Neutrophil apoptosis was studied using our transgenic *TgBAC(mpx:CFP-DEVD-YFP)sh237* (*Robertson et al., 2016*) zebrafish line, which expresses a genetically encoded FRET biosensor consisting of a caspase-3 cleavable DEVD sequence flanked by a CFP YFP pair (*Tyas et al., 2000*), under the neutrophil-specific mpx promoter. A loss of FRET signal in this system provides a read out of apoptosis specifically in neutrophils in vivo in real time. To visualise apoptotic events in the context of neutrophil swarming, 3 dpf *TgBAC(mpx:CFP-DEVD-YFP)sh237* larvae were injured and mounted in a 1% agarose solution containing 0.168 mg/ml tricaine and covered with 500 µl of a clear E3 solution containing tricaine to prevent dehydration. FRET imaging was performed from 30 min postinjury for 5 hr using a 20× objective lens on an UltraVIEWVoX spinning disk confocal laser imaging system (Perkin Elmer) with acquisition every 2 min. Ten z-planes were captured per larvae over a focal range of 100 µm using the following filters: a donor CFP channel (440 nm for excitation, 485 nm for detection), an acceptor YFP channel (514 nm for excitation and 587 nm for detection), and a FRET channel (440 nm for excitation and 587 nm for detection). An Ultraview dichroic mirror passes 405,440,515,640 was used to increase imaging speed using these filter blocks. Volocity software was used to calculate normalised FRET values (nFRET). To compensate for the bleed through of the CFP and YFP fluorophores into the FRET channel, FRET bleed through constants were calculated. Control samples containing HeLa cells transfected with CFP alone or YFP alone were imaged using the same settings used for data acquisition of the mpx:FRET zebrafish reporter line. ROIs were drawn around a population of cells in the frame and Volocity software calculated FRET bleed through values as the mean intensity of the recipient channel (FRET) divided by the mean intensity of the source (CFP or YFP). These FRET constants were then used by Volocity to calculate a normalised FRET value. Neutrophil apoptosis was observed by overlaying the YFP and nFRET channels.

## Generation of histone transgenic reporter line

For the generation of H2A transgenic reporter line, the Gateway cloning toolkit was used to generate a construct, where by the neutrophil-specific lyzozyme C promoter (*lyz*) drives mCherry expression fused to the zebrafish H2az2a protein. The gateway components used to assemble this were a 5' vector p5E-MCS lyz containing 6.6 kb of the lysozyme C promoter (*Kwan et al., 2007*), a middle entry vector pME-h2a-mCherry containing zebrafish histone H2az2a fused to mCherry, and a 3' vector containing a polyadenylation site p3E-polyA. The final construct containing Tol2 arms and green heart marker (*cmlc2*:GFP) for easy recognition of successful transgenesis was created by an LR reaction combining the three vectors with the destination vector pDestTol2CG2. The final construct was microinjected (50 ng/µl), along with tol2 transposase mRNA (50 ng/µL), into one-cell stage embryos of the transgenic line *TgBAC(mpx:GFP)i114*. A stable double transgenic line *TgBAC(mpx:GFP)i114; Tg(lyz:h2az2a-mCherry,cmlc2:GFP)sh530* was generated.

## Kidney neutrophil preparation

Kidney(s) from adult zebrafish aged 3–6 months were dissected as previously described (*Palić et al., 2007*; *Lieschke et al., 2001*), pooled in HBSS, homogenised, and pelleted by centrifugation (250 g, 15 min). Pellets were gently resuspended in 6 ml HBSS and layered on 2 ml of lymphocyte

separation medium 1078 (Mediatech; CellGro, AK) in a 15 ml falcon tube and centrifuged (400 g, 30 min). The resulting layer of leukocytes was removed with a 1 ml sterile pipette and transferred to a 15 ml tube. HBSS was added to a total volume of 4 ml and leukocytes collected by centrifugation (400 g, 15 min). The leukocyte pellet was resuspended in 1 ml HBSS/kidney, and cell yield assessed using a haemocytometer. Preparations yielded $1.1 \pm 0.6 \times 10^6$ cells (n = 15 independent preparations) and were $88.7 \pm 6.2\%$ pure (n = 9 random fields).

### In vitro NET release assay

Two hundred microlitre purified kidney neutrophils from *Tg(lyz:dsRed)nz50* zebrafish cultured in HBSS were plated and left for 30 min to adhere before stimulation with phorbol myristate acetate (PMA) or calcium ionophore solutions containing SYTOX-green. SYTOX-green (Molecular Probes, Eugene) was prepared immediately before use. Stocks were diluted in HBSS for each assay. Final concentrations in assays were as follows: calcium ionophore 100 µg/ml, PMA 10 µg/ml, and SYTOX-green 1 µM. As microbial stimuli, preparations of *S. aureus* and *C. albicans* were used (generous gift of Dr A. Peleg, Monash University). Cells were mounted under coverslips for imaging at time points indicated in figures up to 120 min post-stimulation.

### Staining and immunohistochemistry

Propidium iodide: Live 3 dpf *mpx:GFP* larvae were incubated in 1% LMP agarose solution containing 0.1% propidium iodide (Sigma-Aldrich) immediately following tail fin transection. Pearson's colocalisation analysis was performed by drawing a region of interest around neutrophil cytoplasmic vesicles using Volocity software. DAPI: For DNA staining of sh530 larvae, 2 dpf larvae were fixed in 1 ml of 4% paraformaldehyde (PFA) at room temperature for 30 min, washed in PBST, and transferred to 100% MeOH overnight at −20°C. Samples were washed in PBST twice before permeabilisation using proteinase K (10 µg/ml) for 20 min at room temperature. Samples were fixed for 20 min in 4% PFA at room temperature and washed twice in PBST. Samples were stained in a 0.1% DAPI (Sigma Aldrich) solution in 1× PBS for 20 min and kept in the dark. Samples were washed in PBST, and imaging was performed. Immunohistochemistry: primary antibodies were rabbit anti mpx 1:200 (GeneTex 128379) and chicken anti-EGFP 1:2000 (Abcam ab13970). Secondary antibodies were used 1:1000 from Jackson ImmunoResearch (goat anti-rabbit Alexafluor 647 and goat anti-chicken Alexafluor 488). For antibody staining, paraformaldehyde-fixed (2%) whole embryos were washed twice in PBS-Tx for 30 min and incubated with primary antibody overnight at 4°C. Samples were washed three times in PBS-Tx for 30 min and incubated at 4°C overnight in secondary antibodies. Samples were washed (PBS-Tx, 30 min, three times; PBS, 10 min once) and imaged on Zeiss LSM710 and Leica SP5 microscopes.

### Photoconversion of endogenous pioneer neutrophils

Photoconversion assays were performed using larvae expressing the photoconvertible protein Kaede under the neutrophil-specific *mpx* promoter: *TgBAC(mpx:GAL4-VP16)sh256; Tg(UAS:Kaede)s1999t* (*Elks et al., 2011*). At 3 dpf larvae were anaesthetised and injured using the minor tail fin nick and mounted immediately in a 1% LMP agarose solution containing tricaine. At 10 min post-injury, a region of interest was drawn around the neutrophil nearest to the injury site for photoconversion from green to red fluorescence. Photoconverting of Kaede-labelled neutrophils at the wound site was performed using an UltraVIEWPhotoKinesis device (Perkin Elmer and Analytical Sciences) on an UltraVIEWVoX spinning disk confocal laser imaging system (Perkin Elmer). The photokinesis device was calibrated using a coverslip covered in photobleachable substrate (Stabilo Boss, Berks, UK). Photoconversion was performed using a 405 nm laser at 40% using 120 cycles, 250 pk cycles, and 100 ms as previously published (*Elks et al., 2011*). Successful photoconversion was detected through loss of emission detected following excitation at 488 nm and gain of emission following 561 nm excitation. Following photoconversion, time lapse imaging was performed from 20 min post-injury for 4 hr. Photoconverted neutrophils that became swarm-initiating pioneer neutrophils were analysed.

### Caspase inhibition

Tailfin transection was performed on 2dpf *TgBAC(mpx:GFP)i114* zebrafish larvae.

At 2 hpi, larvae without an inflammatory response were excluded. zVAD-fmk (Santa Cruz Biotechnology) was then added to the fish water at a concentration of 100 µM. Neutrophil swarms were assessed at 6 hpi. Treatment groups of inhibitor experiments were blinded to the experimenter until post-analysis.

### Gasdermin D inhibition

1 dpf *TgBAC(mpx:GFP)i114* zebrafish larvae were incubated for 21 hr with LDC7559 (Medchem express) at 10 µM in fish water. Tailfin transections were performed at 2 dpf with neutrophil swarms assessed at 3 hpi.

### Neutrophil elastase inhibition

2 dpf *TgBAC(mpx:GFP)i114* zebrafish larvae were injected with 1 nl of 80 mM MeOSu-AAPV-CMK (Sigma-Aldrich) via the Duct of Cuvier. Tailfin transections were performed 1 hr post-injection, and neutrophil swarms were assessed at 3 hpi.

### ROS inhibition

2 dpf *TgBAC(mpx:GFP)i114* zebrafish larvae were incubated in diphenyleneiodonium chloride (DPI, Sigma-Aldrich) for either 4 hr pre-injury or 1 hr post-injury at 10 µM in fish water. Tailfin transections were performed and neutrophil swarms assessed at 3 hpi, with neutrophil counts at the tailfin performed at 4 hpi.

### Statistical analysis

Data were analysed using GraphPad Prism version 8.0. Unpaired or paired t-tests were used for comparisons between two groups, and one-way or two-way ANOVA with appropriate post-test adjustment was used for comparisons of three or more groups.

## Acknowledgements

The authors would like to thank the Biological Services Aquarium Team at the University of Sheffield for their assistance with zebrafish husbandry. Imaging work was performed at the Wolfson Light Microscopy Facility, microscopy studies were supported by an MRC grant (G0700091) and a Wellcome Trust grant (GR077544AIA). The authors would also like to thank Dr. Ian Patten for his invaluable scientific writing consultancy. We are very grateful to Dr. Tomasz Prajsnar and Professor Anton Peleg for providing bacterial strains. Special thanks to Dr. Simon Johnson and Dr. Iwan Evans for their mentorship and intellectual input throughout the course of this work. This manuscript has been released as a Pre-Print on Biorxiv (*Isles, 2019b*) and is available at https://doi.org/10.1101/521450.

## Additional information

### Funding

| Funder | Grant reference number | Author |
| --- | --- | --- |
| Medical Research Council | MR/M004864/1 | Stephen A Renshaw |
| Medical Research Council | G0700091 | Stephen A Renshaw |
| Biotechnology and Biological Sciences Research Council | B/R015457/1 | Stephen A Renshaw |
| Wellcome Trust | 105570/Z/14/A | Philip M Elks |
| National Health and Medical Research Council | 1044754 | Graham J Lieschke |
| National Health and Medical Research Council | 1086020 | Graham J Lieschke |

The funders had no role in study design, data collection and interpretation, or the decision to submit the work for publication.

## Author contributions
Hannah M Isles, Conceptualization, Data curation, Formal analysis, Investigation, Methodology, Writing - original draft, Writing - review and editing; Catherine A Loynes, Data curation, Formal analysis, Supervision, Investigation, Methodology, Writing - original draft, Writing - review and editing; Sultan Alasmari, Conceptualization, Data curation, Formal analysis, Investigation, Methodology, Writing - original draft; Fu Chuen Kon, Jack Hales, Clare F Muir, Data curation, Formal analysis, Investigation, Methodology; Katherine M Henry, Resources, Data curation, Investigation, Methodology; Anastasia Kadochnikova, Data curation, Software, Formal analysis; Maria-Cristina Keightley, Supervision; Visakan Kadirkamanathan, Software, Supervision; Noémie Hamilton, Supervision, Writing - review and editing; Graham J Lieschke, Conceptualization, Resources, Data curation, Formal analysis, Supervision, Funding acquisition, Investigation, Methodology, Writing - original draft, Project administration, Writing - review and editing; Stephen A Renshaw, Philip M Elks, Conceptualization, Data curation, Formal analysis, Supervision, Funding acquisition, Investigation, Methodology, Writing - original draft, Project administration, Writing - review and editing

## Author ORCIDs
Hannah M Isles ⓘ https://orcid.org/0000-0002-6019-4842
Fu Chuen Kon ⓘ http://orcid.org/0000-0001-9907-914X
Katherine M Henry ⓘ http://orcid.org/0000-0003-0554-2063
Maria-Cristina Keightley ⓘ https://orcid.org/0000-0001-8141-4069
Visakan Kadirkamanathan ⓘ http://orcid.org/0000-0002-4243-2501
Stephen A Renshaw ⓘ https://orcid.org/0000-0003-1790-1641
Philip M Elks ⓘ https://orcid.org/0000-0003-1683-0749

## Ethics
Animal experimentation: All zebrafish were raised in the Biology Services Aquarium (BSA) at the University of Sheffield in UK Home Office approved aquaria or in the FishCore aquarium at Monash University, Melbourne, Australia, and were maintained following standard protocols. Adult fish were maintained at 28°C with a continuous re-circulating water supply and a daily light/dark cycle of 14/10 hours. All procedures were performed on larvae less than 5.2dpf which were therefore outside of the Animals (Scientific Procedures) Act, to standards set by the UK Home Office. Animal experiments performed in Australia conformed to "Australian code for the care and use of animals for scientific purposes (2013)" and were undertaken under protocol MAS/2010/18 approved by the MARP2 Animal Ethics Committee at Monash University.

## Decision letter and Author response
Decision letter https://doi.org/10.7554/eLife.68755.sa1
Author response https://doi.org/10.7554/eLife.68755.sa2

# Additional files

## Supplementary files
• Transparent reporting form

## Data availability
CSV files of numerical data has been provided for graphs in figures 1, 2, 3, 4, 7 and associated figure supplements. The datasets analysed for this study can be found in the Blood Atlas of Single Cells in zebrafish (BASiCz) database (https://www.sanger.ac.uk/science/tools/basicz) and a database of single cell RNA sequencing of larval zebrafish neutrophils, data deposited on GEO under accession number GSE78954 (https://www.ncbi.nlm.nih.gov/geo/query/acc.cgi?acc=GSE78954).

The following previously published datasets were used:

| Author(s) | Year | Dataset title | Dataset URL | Database and Identifier |
|---|---|---|---|---|
| Cvejic A | 2016 | Blood Atlas of Single Cells in Zebrafish (BASiCz) | https://www.sanger.ac.uk/science/tools/basicz | Array Express, (E-MTAB-3947/E-MTAB-4617/E-MTAB-5530/GSE75478) |
| Rougeot J, Zakrzewska A, Torraca V, Kanwal Z, Jansen H, Spaink HP, Meijer AH | 2017 | RNAseq profiling of FACS-sorted zebrafish larval macrophages reveals similarities with human M1 and M2 transcriptome signatures | https://www.ncbi.nlm.nih.gov/geo/query/acc.cgi?acc=GSE78954 | NCBI Gene Expression Omnibus , GSE78954 |

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
