## [Decision Letter]

**Acceptance summary:**

This manuscript describes the different phases of neutrophil swarming response upon infection via live imaging analysis, using a model of zebrafish larvae and a combination of transgenic reporter lines. Importantly, the authors propose the release of NET occurring in pioneer neutrophils as a critical signal for the initiation of the swarming response. Overall, this paper will be of interest to immunologists who study the triggers of activation of the innate response during infection as well as the consequences of lytic cell death on the immune response.

**Decision letter after peer review:**

[Editors’ note: the authors submitted for reconsideration following the decision after peer review. What follows is the decision letter after the first round of review.]

Thank you for submitting your work entitled "Endogenous pioneer neutrophils release extracellular traps during the swarming response" for consideration by *eLife*. Your article has been reviewed by 3 peer reviewers, including Carla V Rothlin as the Reviewing Editor and Reviewer #1, and the evaluation has been overseen by a Senior Editor. Our decision has been reached after consultation between the reviewers. Based on these discussions and the individual reviews below, we regret to inform you that your work will not be considered further for publication in *eLife*.

Although we are not able to publish this paper, in view of the comments regarding the significance of the findings we remain interested in the topic and your work. We would thus be interested in receiving a new paper when you are able to obtain additional data to address fully the concerns raised in the reviews. As you will recognize from the comments below, one of the main criticisms centers on whether NETosis of the pioneer neutrophil indeed drives neutrophils swarming.

*Reviewer #1:*

Isles and colleagues identify pioneer neutrophils as the initiators of neutrophil swarming. Specifically, the authors make use of the zebrafish model to visualize the early events in response to tail fin transection or infection in the otic vesicle. Using this approach, they identify morphological and behavioral changes – round morphology and reduced motility – in the first neutrophil responding to the injury. This precedes neutrophils swarming. Furthermore, the morphological and behavioral changes are not the consequence of apoptosis, as measured by a reporter line. By contrast the authors identify that pioneer neutrophils release fragments that are positive for DNA and Histones, suggesting that the pioneer neutrophils die by NETosis. The authors also confirmed the requirement of LTB4 signaling in neutrophil recruitment at later stages of the response. In summary, the authors propose that death by NETosis of pioneer neutrophil initiates neutrophil swarming. Overall this is a very interesting manuscript that describes the neutrophil response to injury and infection with high resolution as a result of using an ideal and optically transparent organism.

My main suggestion to strengthen the connection between the proposed role of NETosis in initiating neutrophils swarming would be to interfere with NETosis (either using inhibitors or genetic approaches) and test whether the initial changes in response to injury and/or infection are halted or delayed.

*Reviewer #2:*

The study of Isles et al. with the title „Endogenous pioneer neutrophils release extracellular traps during the swarming response" provides the first comprehensive and systematic study of neutrophil swarming in zebrafish larvae. It confirms the multistep nature of neutrophil swarm formation and a functional contribution of the lipid LTB4 during this response, thus confirming earlier findings in the mouse model. This goes beyond earlier studies in zebrafish where the swarm-like behavior of neutrophils has been briefly mentioned in the context of bacterial infections (Deng/Huttenlocher, J Leuk Biol 2013). In the mammalian system a correlation between the death of early recruited neutrophils as an amplifying signal for swarm formation was shown, but the mechanisms behind it remained difficult to address and are still unknown (Lämmermann/Germain, Nature 2013). By using the power of the zebrafish system (less neutrophils, less complex), this study attempted to dissect an underlying mechanism. The authors make two observations: (1) One pioneer neutrophil undergoes shape changes before they appear to become the center of a developing swarm, (2) Swarm centers show signs of lytic neutrophil cell death, some of which likely due to NETosis.

Overall, the study tries to imply that "chemoattractant signals released by pioneer neutrophils undergoing NETosis could drive the swarming response" (Line 232-234) (see also tentative title). This would indeed be an interesting novel aspect of neutrophil swarm formation worth publishing in *eLife*. However, the current data does not provide solid proof to support this hypothesis (see substantive concerns). The authors are probably aware of the limitations in their data sets, as they use very cautious wording for many of their conclusions on if/how NETosis might influence swarm dynamics.

Unfortunately, this study does not provide enough novel mechanistic insight and go beyond earlier observations on neutrophil swarming.

In their impact statement the authors correctly summarize their findings: "Neutrophils swarm around a single pioneer neutrophil in a zebrafish in vivo model of inflammation AND undergo cell death releasing their cellular components in extracellular traps." However, the authors are missing data that clearly functionally link both processes (which would justify publication in *eLife*). If both events occur completely independently from another, this study would be of minor interest for the general readership.

The authors do show that zebrafish neutrophils can undergo NETosis (Figure 8, Suppl. Figure 8). However, a clear cause-effect relationship that NETosis by pioneer neutrophils drives neutrophil swarming could not be proven. There are a number of question marks regarding the current data sets:

1) The authors refer to "neutrophil fragments that appear around developing neutrophil swarm clusters, sometimes accompanied by large cytoplasmic clusters (Figure 7, Suppl. Figure 6, 7). Together with PI stainings, they later suggest that these events reflect NETosis as potential trigger for neutrophil swarming. However, Supplementary Videos 10-12 do not show a clear cause-effect relationship between dying cells and additional recruitment of neutrophils in the course of amplified swarm growth. In all videos, it even appears as if two different forms of cell death are occurring: (a) neutrophils that release fragments and (b) neutrophils that instantly swell and burst. Only the swelling cells incorporate PI as sign of NETosis (Video 11), but do not appear to become the center of a developing swarm (all videos). Moreover, not only the first (pioneer) neutrophils seem to undergo a lytic form of cell death, but also following (scouting) neutrophils – how can the authors distinguish between the effects of cell death in one or the other neutrophil population? From the data and videos provided it seems that swarming neutrophils in general undergo NETosis, even after initiation of the swarming process.

2) The authors have a strong bias toward suggesting a functional role of NETosis for neutrophil swarming. Necrosis and most forms of NETosis are lytic forms of cell death that could release cellular contents with potential attractive function for neutrophils. The additional release of extracellular DNA (the hallmark of NETosis) may only be a side-effect and not contribute at all to any chemotactic processes during swarming. I would even expect that the disruption of extracellular DNA (by DNAseI treatment, gold standard method to prove NETs in mammals) does not influence swarming. Can the authors rigorously distinguish between necrotic and NETotic death in their system? If not, the authors should tone down many of their statements on NETosis and refer more general to lytic cell death. In mammalian cells, the major (but not exclusive) mechanism to induce NET formation is through NADPH-oxidase and can be blocked by ROS scavenging or NADPH oxidase deficiencies. Would functional interference in a similar manner influence neutrophil swarming in zebrafish larvae?

3) As another method to show neutrophil death by NETosis, the authors use transgenic zebrafish line with fluorescently tagged histones. In Figure 9 and 10, they refer to a catapult-like release of NETs or of histones from the nucleus, but do not show a convincing accompanying video. Moreover, it remains unclear if these were neutrophils in the middle of a neutrophil cluster. Is this histone release also seen with pioneer neutrophils? If so, how many pioneer neutrophils would show such NET release? And how is this then correlated with the recruitment of novel neutrophils and amplification of swarm growth (Video missing)? This is in line with point1 showing a clear cause-effect relation between dying pioneer neutrophils and additional recruitment of neutrophils.

4) In the mammalian system also "viable NETosis" has been observed (Yipp/Kubes, Nat Med 2012); neutrophils stay alive while expelling their extracellular DNA. From the presented data in this manuscript one would argue that this "silent" form of NETosis should not lead to swarming? Do they authors observe this?

5) The authors show that bacterial infections also cause neutrophil swarms even more prominent than during sterile tissue injury. Which neutrophils are dying in the bacteria infection model? Is NETosis also seen with pioneer neutrophils (and other neutrophils)? How does cell death influence this swarming response?

*Reviewer #3:*

In the following manuscript, Isles HM and colleagues examine the neutrophil swarming response upon infection using a model of zebrafish larvae. By tracking neutrophils over time in various transgenic zebrafish models, the authors clearly described the different phases of neutrophil swarming response. Additionally, they propose NETosis occurring in pioneer neutrophils within developing swarms as potential critical signal for regulation and/or development of the swarming response.

In general, the manuscript provides an appropriately balanced interpretation of the findings reported and the study is overall clearly and well presented.

However,

1. the novelty of this contribution is somewhat limited since most of the conclusions reported have been already described in mouse experimental models (The dependency of neutrophils swarming from LTB4 signalling – Lammermann T et al. 2013, the neutrophil swarm upon *S. aureus* infection – Ania Bogoslowski et al., PNAS 2018, the different phases of neutrophil swarming – Ng LG et al., 2011, the distinction between pioneer and scouting neutrophils – Chtanova T et al., 2008, as well as the presence of cell death in few neutrophils at the damage site and the onset of the second phase of the swarming response – Lammermann T et al. 2013) and, additionally, some of the data presented have been already reported in zebrafish models (reviewed in Harvie EA et al., 2015 and Kienle K et al., 2016). All these already published evidences support the idea that neutrophil swarm is a well-defined process, conserved across vertebrates.

On the contrary, their finding on NETosis of an individual "pioneer" neutrophil within the developing swarm, together with all the morphological changes analyzed thanks to the sophisticated live imaging analysis, is of interest and potentially important, although still premature. The data indeed describe for the first time NET formation and neutrophil swarming as connected processes. However, in the actual form, this part lacks convincing functional and mechanistic data and therefore would require further investigations.

2. It would be indeed important to investigate the consequence of NETosis blocking (by inhibition of either elastase, or myeloperoxidase, or PAD4) on neutrophil swarming via in vitro and in vivo experimental assays. Additionally, the identification of the triggers of NETosis in pioneer neutrophils may provide additional insight into the mechanisms leading to swarm regulation.

[Editors’ note: further revisions were suggested prior to acceptance, as described below.]

Thank you for submitting your article "Pioneer neutrophils release chromatin within in vivo swarms" for consideration by *eLife*. Your article has been reviewed by 3 peer reviewers, including Carla V Rothlin as the Reviewing Editor and Reviewer #1, and the evaluation has been overseen by Tadatsugu Taniguchi as the Senior Editor.

Essential revisions:

1) The authors need to relate their findings to the two most closely related studies to this subject (Poplimont/Sarris, Current Biology 2020; Hopke/Irimia, Nature Communications 2020).

2) The authors do not provide a clear link that swarm initiation depends on chromatin release. It rather appears that these events are two separate processes. Unless the authors can show rigorous data that these two events are clearly linked, they have to tone down some of their statements in the manuscript that suggest this functional interrelation.

3) A time-resolved analysis of Figure 7 could further help delineate a role for NET components for early or late phases of swarming.

*Reviewer #1:*

Isles and colleagues identify pioneer neutrophils as the initiators of neutrophil swarming. Specifically, the authors make use of the zebrafish model to visualize the early events in response to tail fin transection or infection in the otic vesicle. Using

this approach, they identify morphological and behavioral changes – round

morphology and reduced motility – in the first neutrophil responding to the injury. This precedes neutrophils swarming. Furthermore, the morphological and behavioral changes are not the consequence of apoptosis, as measured by a reporter line. By contrast, the authors show that pioneer neutrophils release fragments that are positive for DNA and Histones. In summary, the authors propose that death by NETosis of pioneer neutrophil initiates neutrophil swarming. Overall this is a very interesting manuscript that describes the neutrophil response to injury and infection with high resolution as a result of using an ideal and optically transparent organism.

The authors have addressed my prior comments.

*Reviewer #2:*

In this manuscript Isles HM and colleagues described the different phases of neutrophil swarming response upon infection using a combination of transgenic reporter lines in zebrafish larvae. While on the one side many of these data recapitulate in zebrafish the work of others in mouse experimental models (Lammermann T et al. 2013; Ania Bogoslowski et al., PNAS 2018; Ng LG et al., 2011; Chtanova T et al., 2008; Lammermann T et al. 2013), on the other side, here the authors show for the first time in vivo a link between NET release and the neutrophil swarming response. In particular, via the use of inhibitors of NET release and NET formation in vivo, the authors prove the critical role of NET release in pioneer neutrophils as critical trigger of swarm initiation and formation.

The authors have addressed the main concern of this reviewer. The Figure 7 in the new version of the manuscript links NET formation and neutrophil swarming via blocking of neutrophil elastase or MPO. These data provide novel insights into the mechanisms leading to swarm regulation.

*Reviewer #3:*

In the provided manuscript, Isles et al. use transgenic zebrafish larvae to visualize and study the early phases of neutrophil swarm formation. They focus on two observations: (1) neutrophil swarms initiate around individual neutrophils, referred to as pioneer neutrophils, (2) pioneer and early recruited neutrophils release chromatin reminiscent of neutrophil extracellular traps (NET) in neutrophil clusters. The authors propose a functional link between both events with impact on the neutrophil swarming response.

Strengths of this manuscript: The authors convincingly show that zebrafish neutrophils can release extracellular DNA after treatment with known NET stimuli. Genetic and chemical interference with key NET components including gasdermin, neutrophil elastase and myeloperoxidase show a functional requirement of these components in the neutrophil swarming process. The manuscript promotes the release of chromatin from pioneer or early recruited neutrophils as an important event for swarming. They provide live imaging data showing the lytic death of neutrophils as balloon-like structures, including the use of novel transgenic fish lines.

Weaknesses of this manuscript: The authors fail to present their methods in the context of the two most closely related studies to this subject (Poplimont et al., Current Biology 2020; Hopke et al., Nature Communications 2020) and do not cite these studies. Poplimont et al. 2020 (PMID: 32502410) recently revealed a mechanism that pioneer neutrophils undergoing calcium bursts initiate the swarming cascade in the zebrafish model. Dying neutrophils in form of balloon-like structures were also described in detail in this study (Poplimont et al., Video S3 as an example), but could not be correlated to contribute to the initiating events of neutrophil swarm formation. Hopke et al. 2020 (PMID: 32341348) showed an involvement of NET components during swarm formation of human neutrophils in vitro. Interference with NET components in this study (e.g. myeloperoxidase inhibition) did not interfere with initial swarm recruitment, but rather later phases of the neutrophil swarm response. From these two studies, it appears that the two processes of swarm initiation and the potential role of extracellular NET components are separate processes during neutrophil swarm formation. The provided manuscript of Isles et al. does not provide convincing data that clearly link these two events.

Overall, Isles et al. provide a comprehensive study on neutrophil swarming in the zebrafish system. They confirm the multistep nature of the swarming response in zebrafish, which was known for other species, including the involvement of LTB4. Their experiments also make a strong argument for a functional role of extracellular NET components during neutrophil swarm growth. However, it remains unclear from the presented data which phases (early vs. late) of neutrophil swarm growth are controlled by these components. Time-resolved analysis of the findings in Figure 7 could provide answers to this open question.

Unfortunately, the presented data – together with published work (e.g. Poplimont et al. 2020) – do not support a number of statements of the manuscript, e.g. "our findings demonstrate that an individual pioneer neutrophil initiates the swarm and releases chromatin, a process followed by the directed migration of swarming neutrophils." or "we propose that the pioneer neutrophil is able.… to release extracellular DNA that amplifies chemoattraction". The release of chromatin does not always and not exclusively occur only in pioneer neutrophils. The initiation of swarming may involve another mechanism, which likely do not rely on the release of chromatin.

The functional contribution of neutrophil cell death to the swarming response is an important question to the field of inflammation, as it may explain why some neutrophil infiltrates continue growing, while others will shrink again over time. Neutrophil death inducing conditions (e.g. bacteria-induced cell lysis) may further exacerbate swarm growth. As the swarming response is pretty conserved among species, findings from the zebrafish model are likely to be also relevant for inflammatory processes in mammals, including humans.

---

## [Author Response]

[Editors’ note: the authors resubmitted a revised version of the paper for consideration. What follows is the authors’ response to the first round of review.]

Reviewer #1:[…] My main suggestion to strengthen the connection between the proposed role of NETosis in initiating neutrophils swarming would be to interfere with NETosis (either using inhibitors or genetic approaches) and test whether the initial changes in response to injury and/or infection are halted or delayed.

We agree! The mechanistic link between NET release and swarming is really important and something we have been working hard on. We have now addressed this using a range of inhibitors and genetic approaches and believe this has greatly improved the manuscript and now provides the first in vivo mechanistic link between NET release and swarming. We found that inhibition of key players in NET formation leads to a decrease in the occurrence of neutrophil swarming. For Gasdermin D (GSDM) (Sollberger et al., Science Immunology, 2018) and neutrophil elastase (Papayannopoulos et al., J Cell Biol, 2010) we chose an inhibitor approach as the inhibitors are well characterised (and the genetics of the fish orthologues of these are excessively complex to effectively unpick using genetic approaches, outlined on lines 318-326 and 328-342). We found that treatment with the pore blocking gasdermin inhibitor with LDC7559, or treatment with MeOSu-AAPV-CMK to block leukocyte elastase, significantly decreased swarm occurrence, compared to DMSO treated controls (new Figure 7A-B). We also targeted myeloperoxidase, necessary for NET formation (Metzler et al., Blood, 2011). We chose a genetic approach, using CRISPR-Cas9 targeting the myeloperoxidase promoter/start site to knockdown mpx (zebrafish MPO equivalent), showing that this also decreased the level of swarm formation (new figure 7C-D). Taken together, these new data demonstrate a functional link between NET release and swarm formation, not previously demonstrated during in vivo inflammation and adds further novelty and impact to our manuscript.

Reviewer #2:[…] Overall, the study tries to imply that "chemoattractant signals released by pioneer neutrophils undergoing NETosis could drive the swarming response" (Line 232-234) (see also tentative title). This would indeed be an interesting novel aspect of neutrophil swarm formation worth publishing in eLife. However, the current data does not provide solid proof to support this hypothesis (see substantive concerns). The authors are probably aware of the limitations in their data sets, as they use very cautious wording for many of their conclusions on if/how NETosis might influence swarm dynamics.Unfortunately, this study does not provide enough novel mechanistic insight and go beyond earlier observations on neutrophil swarming.

We have now added further mechanistic insights (details below and in comments to reviewer 1). By inhibiting NET components genetically and pharmacologically (in new Figure 7) we have demonstrated a functional link between NET release and neutrophil swarming that we believe represents an important step forwards in our understanding of swarming in vivo. The manuscript has been extensively revised (in response to the helpful suggestions of all the reviewers) to include new mechanistic insight and to better communicate our message and all figures have been remade (again at the helpful suggestion of the reviewers) to improve clarity.

In their impact statement the authors correctly summarize their findings: "Neutrophils swarm around a single pioneer neutrophil in a zebrafish in vivo model of inflammation AND undergo cell death releasing their cellular components in extracellular traps." However, the authors are missing data that clearly functionally link both processes (which would justify publication in eLife). If both events occur completely independently from another, this study would be of minor interest for the general readership.

We have now addressed this concern and mechanistically linked the release of NETs and neutrophil swarming (see response to reviewer one and New Figure 7). We believe this additional mechanistic insight would now justify publication in *eLife*, especially considering that these observations are in vivo, in a whole organism tissue context of inflammation.

The authors do show that zebrafish neutrophils can undergo NETosis (Figure 8, Suppl. Figure 8). However, a clear cause-effect relationship that NETosis by pioneer neutrophils drives neutrophil swarming could not be proven. There are a number of question marks regarding the current data sets:1) The authors refer to "neutrophil fragments that appear around developing neutrophil swarm clusters, sometimes accompanied by large cytoplasmic clusters (Figure 7, Suppl. Figure 6, 7). Together with PI stainings, they later suggest that these events reflect NETosis as potential trigger for neutrophil swarming. However, Supplementary Videos 10-12 do not show a clear cause-effect relationship between dying cells and additional recruitment of neutrophils in the course of amplified swarm growth. In all videos, it even appears as if two different forms of cell death are occurring: (a) neutrophils that release fragments and (b) neutrophils that instantly swell and burst. Only the swelling cells incorporate PI as sign of NETosis (Video 11), but do not appear to become the center of a developing swarm (all videos). Moreover, not only the first (pioneer) neutrophils seem to undergo a lytic form of cell death, but also following (scouting) neutrophils – how can the authors distinguish between the effects of cell death in one or the other neutrophil population? From the data and videos provided it seems that swarming neutrophils in general undergo NETosis, even after initiation of the swarming process.

We have substantially altered the manuscript textually to address this point. We do not use the term NETosis and refer instead to NET release. To address exactly what type of cell death is occurring is extremely technically challenging in vivo due to a lack of available tools. We do however add further mechanistic insight that narrows down what type of lytic death may be occurring. We have added evidence that the pioneer neutrophil is not undergoing caspase dependent cell death, using FRET reporter and Caspase inhibition (in new Figure 3G) which is an important question in the swarm field, especially in vivo. Furthermore we show that Gasdermin D, myeloperoxidase and neutrophil elastase are required (new Figure 7 and see response to reviewer 1 above). We agree with the reviewer that early swarming neutrophils, alongside the pioneer are releasing NETs and have changed the text to make this point more clearly.

2) The authors have a strong bias toward suggesting a functional role of NETosis for neutrophil swarming. Necrosis and most forms of NETosis are lytic forms of cell death that could release cellular contents with potential attractive function for neutrophils. The additional release of extracellular DNA (the hallmark of NETosis) may only be a side-effect and not contribute at all to any chemotactic processes during swarming. I would even expect that the disruption of extracellular DNA (by DNAseI treatment, gold standard method to prove NETs in mammals) does not influence swarming.

We understand the wish to try DNAse 1, the gold-standard method used in mammalian systems, but it is important to point out that it is the in vitro gold standard, not used extensively in in vivo NET research. Zebrafish larvae are amenable to drug treatments, both by addition to the embryo water or by microinjection into the circulation. We attempted to deliver of a range of DNAse 1 concentrations by both methods and found it to be incredibly toxic to the larvae, killing the fish within a few hours. We could therefore not use this in vitro method in vivo. Instead, we focused on disrupting NET components using multiple methods outlined in the response to reviewer one, which we believe is a more elegant approach with fewer off-target effects in vivo.

Can the authors rigorously distinguish between necrotic and NETotic death in their system? If not, the authors should tone down many of their statements on NETosis and refer more general to lytic cell death.

We agree with this point (see response above) and now use the more precise terms NET release and lytic cell death, as suggested by this reviewer.

In mammalian cells, the major (but not exclusive) mechanism to induce NET formation is through NADPH-oxidase and can be blocked by ROS scavenging or NADPH oxidase deficiencies. Would functional interference in a similar manner influence neutrophil swarming in zebrafish larvae?

In mammalian systems ROS can be an important player in the onset of NET forming pathways is NADPH-oxidase induced reactive oxygen species (ROS) release, as suggested by the reviewer. We therefore used the NADPH-oxidase inhibitor Diphenyleneiodonium chloride (DPI) to inhibit ROS production in an attempt to modulate swarming. ROS production has been shown to be critical for the onset on neutrophil recruitment (Niethammer et al., Nature, 2009), and, corroborating this, we showed that DPI treatment from 1 hour before tailfin wound inhibited neutrophil recruitment to the wound at 4hpw. However, if initial recruitment was allowed to happen before treatment with DPI at 1hpw, then neutrophil numbers at 4hpw were no different to control DMSO treated larvae allowing us to assess the effects on swarming directly. We then assessed whether these two DPI treatment windows affected the frequency of swarm formation. Treatment from 1 hour before wounding led to a decrease in swarm formation compared to DMSO controls, however treatment from 1hpw showed no difference in swarm frequency compare to controls. When corrected for total neutrophil numbers at the wound, the frequency of swarming per 100 neutrophils was not significantly altered with either timepoint of DPI treatment. These data indicate that swarming is not manipulated by inhibition of ROS at these timepoints. However, we cannot rule out a time-specific contribution of ROS to the process and as such we have decided not to include this data in the main revised manuscript.

3) As another method to show neutrophil death by NETosis, the authors use transgenic zebrafish line with fluorescently tagged histones. In Figure 9 and 10, they refer to a catapult-like release of NETs or of histones from the nucleus, but do not show a convincing accompanying video. Moreover, it remains unclear if these were neutrophils in the middle of a neutrophil cluster. Is this histone release also seen with pioneer neutrophils? If so, how many pioneer neutrophils would show such NET release? And how is this then correlated with the recruitment of novel neutrophils and amplification of swarm growth (Video missing)? This is in line with point1 showing a clear cause-effect relation between dying pioneer neutrophils and additional recruitment of neutrophils.

We describe the histone release is catapult-like due to existing nomenclature, a term defined from in vitro observations where neutrophils throw out large NETs that are flat in structure due to them being on a glass surface. However, in vivo we believe NETs look quite different, exhibiting a “balloon-like” morphology, a terminology we use in the revised manuscript. We do however believe a more subtle “catapult-like” action is happening, as in in vitro observations but due to the 3D tissue nature of the in vivo biology this is less obvious. The original neutrophil is very hard to image as, in order to identify the pioneer neutrophil, we have to look retrospectively once it has formed a swarm, after which it is quickly surrounded by swarming neutrophils shielding it from imaging. We have addressed in Figure 4F that the pioneer neutrophil releases cytoplasmic balloons, however, the Kaede transgenic line technology is not compatible with the histone transgenic due to an overlap in red fluorescence. The number of neutrophils releasing NETs is likely to be variable across different swarms and the packed nature of early swarms make imaging accurately counting individual neutrophils within the swarm technically extremely challenging. In the new manuscript we propose that more than one pioneer neutrophil releases the NETs into early swarms, as suggested by this reviewer.

4) In the mammalian system also "viable NETosis" has been observed (Yipp/Kubes, Nat Med 2012); neutrophils stay alive while expelling their extracellular DNA. From the presented data in this manuscript one would argue that this "silent" form of NETosis should not lead to swarming? Do they authors observe this?

We have not seen this type of NETosis in vivo. It appears to us that lytic cell death is occurring, because we lose fluorescence from the original cell body and the end product is neutrophil cell debris (new Figure 4D). This is highly characteristic of lytic death and we have no evidence that suggests that the cell body remains alive.

5) The authors show that bacterial infections also cause neutrophil swarms even more prominent than during sterile tissue injury. Which neutrophils are dying in the bacteria infection model? Is NETosis also seen with pioneer neutrophils (and other neutrophils)? How does cell death influence this swarming response?

The focus of this manuscript is inflammatory neutrophils at a sterile tailfin wound. The infection data is included for completeness, as part of descriptive work on swarms. We would like to follow up on infection observations in future studies, but believe this to be out of the scope of the current manuscript.

Reviewer #3:[…] 1. the novelty of this contribution is somewhat limited since most of the conclusions reported have been already described in mouse experimental models (The dependency of neutrophils swarming from LTB4 signalling – Lammermann T et al. 2013, the neutrophil swarm upon *S. aureus* infection – Ania Bogoslowski et al., PNAS 2018, the different phases of neutrophil swarming – Ng LG et al., 2011, the distinction between pioneer and scouting neutrophils – Chtanova T et al., 2008 as well as the presence of cell death in few neutrophils at the damage site and the onset of the second phase of the swarming response – Lammermann T et al. 2013) and, additionally, some of the data presented have been already reported in zebrafish models (reviewed in Harvie EA et al., 2015 and Kienle K et al., 2016). All these already published evidences support the idea that neutrophil swarm is a well-defined process, conserved across vertebrates.

We are pleased that the reviewer has recognised the thorough characterisation of swarming in this model and highlights the many areas where our model recapitulates the work of others. We felt this was important before making novel observations on the nature of the pioneer cell death. We have now added further substantial novelty to our manuscript by demonstrating, for the first time in vivo, a mechanistic link between NET release and neutrophil swarming. We believe throughout, that our manuscript demonstrates significant novelty, especially in our observations regarding pioneer neutrophils. Our observations with endogenous neutrophils in their natural tissue environment is a particular benefit of the zebrafish model. The process of swarming in zebrafish, although mentioned incidentally in some recent manuscripts has not been described in detail, and we believe this is the first account that confirms that the process is mechanistically similar (ie, LTB4 dependent, to infection and injury and involving lytic cell death). The comparison to existing murine data is a strength of our manuscript and important to ascertain before our novel findings on the endogenous pioneers and mechanistic links to NET release. Furthermore, we describe here a new zebrafish transgenic line labelling histones that will be of use to the zebrafish field to further develop investigations into NET release and swarming that have thus far been limited due to lack of available tools.

On the contrary, their finding on NETosis of an individual "pioneer" neutrophil within the developing swarm, together with all the morphological changes analyzed thanks to the sophisticated live imaging analysis, is of interest and potentially important, although still premature. The data indeed describe for the first time NET formation and neutrophil swarming as connected processes. However, in the actual form, this part lacks convincing functional and mechanistic data and therefore would require further investigations.

We agree with the reviewer that these observations have potentially important implications. We have added mechanism to these observations as requested (new Figure 7 and see response to reviewer 1).

2. It would be indeed important to investigate the consequence of NETosis blocking (by inhibition of either elastase, or myeloperoxidase, or PAD4) on neutrophil swarming via in vitro and in vivo experimental assays. Additionally, the identification of the triggers of NETosis in pioneer neutrophils may provide additional insight into the mechanisms leading to swarm regulation.

We have now addressed this point to add further in vivo mechanistic insight to our manuscript (see first response to reviewer 1). In the new manuscript we address that inhibition of neutrophil elastase or myeloperoxidase inhibition decreases swarm formation in vivo. On the point of PAD4, the situation in zebrafish is less clear as the exact homologue is not well established. The gene *padi2* has been identified as the putative homologue for human PADI 2 and 4 in zebrafish, and has recently been shown to play key roles in inflammation at the tailfin wound (Goenberg et al., 2020). To modulate Padi2 we used the pan-PAD inhibitor Clamidine that irreversibly inactivates PADs (Silvestre-Roig et al., 2019). Cl^-^amidine was found to significantly increase the frequency of zebrafish with swarms. This finding is contradictory in terms of the predicted roles of Padi2 in neutrophil swarming, but correlates with the recent finding in zebrafish that *padi2* knockout increases neutrophil numbers by increased recruitment to the tailfin wound (Golenberg et al., J Cell Biol, 2020). We were able to recapitulate this increased recruitment using Cl^-^amide, which we believe explains why the swarming rate is increased and, due to this unrelated mechanism, we have not included these data in the new manuscript. The data from ourselves and Golenberg et al., indicate a role outside of NET formation for *padi2* in zebrafish neutrophils, illustrating that the effects of citrullination on neutrophilic inflammation in vivo may be more complex than anticipated from in vitro observations.

[Editors’ note: what follows is the authors’ response to the second round of review.]

Essential revisions:1) The authors need to relate their findings to the two most closely related studies to this subject (Poplimont/Sarris, Current Biology 2020; Hopke/Irimia, Nature Communications 2020).

We apologise for this oversight during the redrafting process and agree this is very important to address. We have now added sections in the introduction (lines 83-90, 98-99, 107-111), results (lines 181-183, 357-359) and discussion (431-435, 437-440, 447-449, 484-526) to relate our findings to Poplimont/Sarris, Current Biology 2020; Hopke/Irimia, Nature Communications 2020 and the even more recent findings from Chen/Yang, JI , 2021 (line 340-341). Furthermore we have now included our DPI data to the main manuscript (new Figure 7—figure supplement 1) so it can be compared to the Hopke/Irimia Nature Comms 2020 study where DPI is used. We observed no reduction in swarming frequency after DPI treatment, which is consistent with Hopke/Irimia’s observations that show that neutrophil swarms rapidly form in the presence of DPI on their in vitro arrays.

2) The authors do not provide a clear link that swarm initiation depends on chromatin release. It rather appears that these events are two separate processes. Unless the authors can show rigorous data that these two events are clearly linked, they have to tone down some of their statements in the manuscript that suggest this functional interrelation.

We thank the reviewers for this comment. We believe our new data confirm the mechanistic link between swarm formation and NET release, although we agree that it is not necessarily the chromatin itself which is responsible. Our data in figure 7 show that targeting of a number of different proteins related to the release of chromatin/NETs leads to a decrease in swarm frequency, thereby confirming functional linkage between the two processes. We believe this to be strong evidence of a functional link between NET release and swarming using multiple approaches and strands of evidence. We do however concede that, unlike the Hopke/Irimia in vitro study, we have not temporally addressed the effects of these compounds on swarm initiation versus later swarm processes. Due to the complex and stochastic nature of swarm formation in vivo we believe this to be out of the scope of the current study. We have changed our statements on chromatin/NET release via the proteins tested (Gasdermin D, elastase and myeloperoxidase) being responsible for “swarm formation” to “swarming” and toned down some of our stronger statements in response to this comment. The Hopke/Irimia study adds in vitro support for our findings of a direct role for NET release in the modulation of swarming. Further elucidation of these mechanisms will require investigation using complementary models and approaches outwith the scope of the current study.

3) A time-resolved analysis of Figure 7 could further help delineate a role for NET components for early or late phases of swarming.

We agree that there remains much to understand about NET component release and swarming. Resolving their roles in early and later swarming processes is one of these questions, as highlighted in the previous comment and by reviewer #3. To fully delineate the roles of NET components in early or late swarming we believe that an integrated approach using a combination of complementary models (eg in vivo zebrafish work alongside in vitro swarming approaches) are required. In our model, we have no control over the initiation of swarms, unlike some other models. To address this question would require the development of techniques to control swarm formation, which is outside the scope of this manuscript.

There is a window of between around 3-6 hours post injury where swarms form and, due to the stochastic nature of when these form within this window (with around 50% of larva not forming swarms at all), further temporal dissection is technically challenging in vivo. We cannot perform timelapse microscopy on the large number of fish required to detect differences in swarming with all the treatment groups and appropriate controls. To see statistically significant differences, we estimate we would need to timelapse at least 100 fish per treatment group and, even then, the range of neutrophil swarming start/end times would make the early vs later events difficult to delineate and compare between groups. This is a major strength of the in vitro models of Hopke/Irimia which allow interrogation of a large number of synchronised swarms, something not possible in zebrafish models currently. An advantage of our zebrafish models is that we can assess the roles of NET forming proteins in the in vivo microenvironment, despite not being able to resolve the timing of events further. We do not believe that we can time resolve to a greater level than this with confidence in the zebrafish model alone and would need to develop new models to delineate timings further in future studies. We have added this to our discussion.

Reviewer #3:[…] Weaknesses of this manuscript: The authors fail to present their methods in the context of the two most closely related studies to this subject (Poplimont et al., Current Biology 2020; Hopke et al., Nature Communications 2020) and do not cite these studies. Poplimont et al. 2020 (PMID: 32502410) recently revealed a mechanism that pioneer neutrophils undergoing calcium bursts initiate the swarming cascade in the zebrafish model. Dying neutrophils in form of balloon-like structures were also described in detail in this study (Poplimont et al., Video S3 as an example), but could not be correlated to contribute to the initiating events of neutrophil swarm formation. Hopke et al. 2020 (PMID: 32341348) showed an involvement of NET components during swarm formation of human neutrophils in vitro. Interference with NET components in this study (e.g. myeloperoxidase inhibition) did not interfere with initial swarm recruitment, but rather later phases of the neutrophil swarm response. From these two studies, it appears that the two processes of swarm initiation and the potential role of extracellular NET components are separate processes during neutrophil swarm formation. The provided manuscript of Isles et al. does not provide convincing data that clearly link these two events.

We mentioned the following in response to Essential Revision #1 but we include our response here as well for convenience. We apologise for this oversight during the redrafting process and agree this is very important to address. We have now added sections in the introduction (lines 83-90, 98-99, 107-111), results (lines 181-183, 357-359) and discussion (431-435, 437-440, 447-449, 484-526) to relate our findings to Poplimont/Sarris, Current Biology 2020; Hopke/Irimia, Nature Communications 2020 and the even more recent findings from Chen/Yang, JI , 2021 (line 340-341). Furthermore we have now included our DPI data to the main manuscript (new Figure 7—figure supplement 1) so it can be compared to the Hopke/Irimia Nature Comms 2020 study where DPI is used. We observed no reduction in swarming frequency after DPI treatment, which is consistent with Hopke/Irimia’s observations that show that neutrophil swarms rapidly form in the presence of DPI on their in vitro arrays.

Overall, Isles et al. provide a comprehensive study on neutrophil swarming in the zebrafish system. They confirm the multistep nature of the swarming response in zebrafish, which was known for other species, including the involvement of LTB4. Their experiments also make a strong argument for a functional role of extracellular NET components during neutrophil swarm growth. However, it remains unclear from the presented data which phases (early vs. late) of neutrophil swarm growth are controlled by these components. Time-resolved analysis of the findings in Figure 7 could provide answers to this open question.

We mentioned the following in response to Essential Revision #3 but we include our response here as well for convenience. We agree that there remains much to understand about NET component release and swarming. Resolving their roles in early and later swarming processes is one of these questions, as highlighted in the previous comment and by reviewer #3. To fully delineate the roles of NET components in early or late swarming we believe that an integrated approach using a combination of complementary models (eg in vivo zebrafish work alongside in vitro swarming approaches) are required. In our model, we have no control over the initiation of swarms, unlike some other models. To address this question would require the development of techniques to control swarm formation, which is outside the scope of this manuscript.

There is a window of between around 3-6 hours post injury where swarms form and, due to the stochastic nature of when these form within this window (with around 50% of larva not forming swarms at all), further temporal dissection is technically challenging in vivo. We cannot perform timelapse microscopy on the large number of fish required to detect differences in swarming with all the treatment groups and appropriate controls. To see statistically significant differences, we estimate we would need to timelapse at least 100 fish per treatment group and, even then, the range of neutrophil swarming start/end times would make the early vs later events difficult to delineate and compare between groups. This is a major strength of the in vitro models of Hopke/Irimia which allow interrogation of a large number of synchronised swarms, something not possible in zebrafish models currently. An advantage of our zebrafish models is that we can assess the roles of NET forming proteins in the in vivo microenvironment, despite not being able to resolve the timing of events further. We do not believe that we can time resolve to a greater level than this with confidence in the zebrafish model alone and would need to develop new models to delineate timings further in future studies. We have added this to our discussion.

Unfortunately, the presented data – together with published work (e.g. Poplimont et al. 2020) – do not support a number of statements of the manuscript, e.g. "our findings demonstrate that an individual pioneer neutrophil initiates the swarm and releases chromatin, a process followed by the directed migration of swarming neutrophils." or "we propose that the pioneer neutrophil is able.… to release extracellular DNA that amplifies chemoattraction". The release of chromatin does not always and not exclusively occur only in pioneer neutrophils. The initiation of swarming may involve another mechanism, which likely do not rely on the release of chromatin.

We mentioned some of the following in response to Essential Revision #2 but we include our response here as well for convenience. We do not believe the findings we report here are mutually exclusive to the findings reported in Poplimont et al., 2020. It is likely that swarming neutrophils are in the process of responding to DAMPs released by the wound as part of the recruitment process (possibly including ATP from necrotic tissue, as indicated in Poplimont et al., 2020, though we do not address this here) but this does not preclude that factors released by pioneer neutrophils are also important in the swarming response. The linear wound produced by tailfin transection in our manuscript is larger than the laser wound performed in Poplimont et al., 2020. The linear nature of the wound led us to the observation that swarms form at any point on the ventral/dorsal axis along the long linear tailfin wound, but always form around a pioneer neutrophil, strongly indicative that signals from pioneer neutrophils are playing an important role in swarming in this model, while not excluding that necrotic signals from the wound itself may also play a role. We believe our new data confirm the mechanistic link between swarm formation and NET release, although we agree that it is not necessarily the chromatin itself which is responsible, and have toned down the statements highlighted by the reviewer. Our data in figure 7 show that targeting of a number of different proteins related to the release of chromatin/NETs leads to a decrease in swarm frequency, thereby confirming functional linkage between the two processes. We believe this to be strong evidence of a functional link between NET release and swarming using multiple approaches and strands of evidence. We do however concede that, unlike the Hopke/Irimia in vitro study, we have not temporally addressed the effects of these compounds on swarm initiation versus later swarm processes. Due to the complex and stochastic nature of swarm formation in vivo we believe this to be out of the scope of the current study. We have changed our statements on chromatin/NET release via the proteins tested (Gasdermin D, elastase and myeloperoxidase) being responsible for “swarm formation” to “swarming” and toned down some of our stronger statements in response to this comment. The Hopke/Irimia study adds in vitro support for our findings of a direct role for NET release in the modulation of swarming. Further elucidation of these mechanisms will require investigation using complementary models and approaches outwith the scope of the current study.

The functional contribution of neutrophil cell death to the swarming response is an important question to the field of inflammation, as it may explain why some neutrophil infiltrates continue growing, while others will shrink again over time. Neutrophil death inducing conditions (e.g. bacteria-induced cell lysis) may further exacerbate swarm growth. As the swarming response is pretty conserved among species, findings from the zebrafish model are likely to be also relevant for inflammatory processes in mammals, including humans.

We would like to thank the reviewer for their insightful comments during the revision process. We are very much in agreement that neutrophil swarming is an exciting and emerging process in the inflammation and infections fields, and integration of the zebrafish model into our understanding of the process will add to our in vivo knowledge of the cellular and molecular mechanisms involved.